# Chitosomes Loaded with Docetaxel as a Promising Drug Delivery System to Laryngeal Cancer Cells: An In Vitro Cytotoxic Study

**DOI:** 10.3390/ijms24129902

**Published:** 2023-06-08

**Authors:** Christian R. Moya-Garcia, Nicole Y. K. Li-Jessen, Maryam Tabrizian

**Affiliations:** 1Department of Biomedical Engineering, Faculty of Medicine and Health Sciences, McGill University, 3775 Rue University, Montreal, QC H3A 2B4, Canada; christian.moyagarcia@mail.mcgill.ca; 2School of Communication Sciences and Disorders, McGill University, 2001 Av. McGill College #8, Montréal, QC H3A 1G1, Canada; 3Department of Otolaryngology—Head and Neck Surgery, McGill University Health Centre, 1001 Decarie Blvd., Montreal, QC H4A 3J1, Canada; 4Research Institute of the McGill University Health Centre, 1001 Decarie Blvd., Montreal, QC H4A 3J1, Canada; 5Faculty of Dental Medicine and Oral Health Sciences, McGill University, 2001 Av. McGill College, Montreal, QC H3A 1G1, Canada

**Keywords:** liposomes, drug delivery, docetaxel, chitosan coating, laryngeal cancer

## Abstract

Current delivery of chemotherapy, either intra-venous or intra-arterial, remains suboptimal for patients with head and neck tumors. The free form of chemotherapy drugs, such as docetaxel, has non-specific tissue targeting and poor solubility in blood that deters treatment efficacy. Upon reaching the tumors, these drugs can also be easily washed away by the interstitial fluids. Liposomes have been used as nanocarriers to enhance docetaxel bioavailability. However, they are affected by potential interstitial dislodging due to insufficient intratumoral permeability and retention capabilities. Here, we developed and characterized docetaxel-loaded anionic nanoliposomes coated with a layer of mucoadhesive chitosan (chitosomes) for the application of chemotherapy drug delivery. The anionic liposomes were 99.4 ± 1.5 nm in diameter with a zeta potential of −26 ± 2.0 mV. The chitosan coating increased the liposome size to 120 ± 2.2 nm and the surface charge to 24.8 ± 2.6 mV. Chitosome formation was confirmed via FTIR spectroscopy and mucoadhesive analysis with anionic mucin dispersions. Blank liposomes and chitosomes showed no cytotoxic effect on human laryngeal stromal and cancer cells. Chitosomes were also internalized into the cytoplasm of human laryngeal cancer cells, indicating effective nanocarrier delivery. A higher cytotoxicity (*p* < 0.05) of docetaxel-loaded chitosomes towards human laryngeal cancer cells was observed compared to human stromal cells and control treatments. No hemolytic effect was observed on human red blood cells after a 3 h exposure, proving the proposed intra-arterial administration. Our in vitro results supported the potential of docetaxel-loaded chitosomes for locoregional chemotherapy delivery to laryngeal cancer cells.

## 1. Introduction

Head and neck cancer is one of the most aggressive cancers [1]. It is considered the sixth most common type of cancer [2], with more than 650,000 newly diagnosed head and neck cancer cases each year worldwide [3]. As a sub-type of head and neck cancer, laryngeal cancer has an annual incidence rate over 30% of the total number of head and neck cancers worldwide, second to oral cancer [4]. The 5-year survival rate of human papilloma virus-negative laryngeal cancer has been approximately 50% for the past three decades [1,5]. 

Induction chemotherapy is the standard-of-care treatment for head and neck cancer [6,7]. In particular, docetaxel (DTX) is one common chemotherapy drug in the treatment of head and neck cancer [8,9,10] that is delivered via systemic intravenous [6,7,11,12,13] or locoregional intra-arterial [14,15,16] routes. Either delivery route has its own pros and cons that undermine the expected benefits of chemotherapy. Systemic intravenous delivery provokes highly toxic deleterious effects throughout the body because the chemotherapy concentration is similar in the tumor site as that present in the whole body [6,7,11,12,13]. Remarkably, about 1% of the chemotherapeutics reach solid tumors via the systemic route [17,18].

Alternatively, locoregional intra-arterial chemotherapy was proposed to overcome such systemic toxicities by infusing the drug into tumor-supplying arteries, rather than circulating the chemotherapeutics systemically [14,15,16]. However, the intra-arterial delivery may still cause toxic extravasation damage in the surrounding tumor region [15,16]. New advances in immunotherapy have been proposed to directly inject a gelatin biomaterial loaded with immune checkpoint blockade drugs into the tumor [19]. However, considering the unique structure and immune environment of the upper airway, only a very small volume of drugs can be injected without causing breathing obstruction [20,21,22]. As such, reducing the unwanted toxic damage from either intra-venous or intra-arterial induction chemotherapy remains a major challenge in head and neck cancer treatment. 

Nanocarriers have been proposed to protect chemotherapy drugs from early degradation and promote targeted delivery to the local tumor [23,24,25]. Among these nanocarriers, liposomes are FDA-approved drug vesicles with decreased dose concentrations providing a controlled and sustained drug release [25]. Liposomes are made of phospholipids and cholesterol [25,26] and have a similar phospholipid bilayer membrane as that of cells. Liposomes have been shown to protect lipophilic/water insoluble drugs, e.g., chemotherapy drugs such as DTX, from rapid degradation while they circulate the blood stream [26,27,28,29]. This feature can help the chemo drug to reach the tumor vasculature and reduce systemic toxicity [30,31,32] (Figure 1).

The physiochemical properties of liposomes can be tuned and tailored to specific drug delivery systems and tumor organs. Liposomes with a diameter between 100 and 200 nm have shown increased intratumoral retention [17,30,31,32,33,34,35]. These sizes allow liposomes to extravasate and disperse, promoting passive targeting in solid tumors as those found in the larynx, due to the enhanced permeability and retention effect [17,30,31,32,33]. The vasculature in the tumor core has an aberrant conformation with irregular fenestrations. The gap junctions of the intratumoral vasculature are widely varied in size between 380 and 780 nm [34], compared to the normal epithelia/endothelia with regular gap junctions (6–12 nm) [36]. This aberrant vasculature may create a dead end for liposome and macromolecule accumulation allowing for uptake by tumor cells without causing damage to non-fenestrated tissues as free drugs do [36,37] (Figure 1).

The surface of liposomes can also be modified by bioadhesive coatings to improve their physical and colloidal stability, bioavailability, and drug entrapment [23,24,38,39,40,41,42,43,44]. For instance, there are high interstitial intratumoral pressures hindering non-adhesive nanocarrier uptake [17]; the application of a chitosan coating on anionic nanoliposomes can serve as a mucoadhesive agent, enhancing the retention of drugs and increase their uptake by target tumors [23,24,38,40,42,45]. Chitosan-coated nanocarriers have shown to attach to anionic glycoproteins in oral mucin [45,46] via electrostatic interactions and, as a result, increased the intratumoral retention rate of the nanocarriers (Figure 1b). This mucoadhesive feature is particularly desired for mucin-dominant mucosae like those commonly found in head and neck tumors. Chitosan-coated nanocarriers have also been shown to have a lower aggregation in the blood and the liver [39], as well as increased transcellular and paracellular drug transport for prolonged drug release [45].

Multiple steps are implemented to synthesize DTX-loaded “chitosomes”, i.e., liposomes coated with chitosan [38,47,48,49]. Lipid components are first dissolved in organic solvents such as ethanol [49] or chloroform [38,47,48] with an extra step to load DTX. The use of ethanol as the organic solvent in liposome synthesis has been reported to increase the reproducibility of liposome particle size and polydispersity index compared to those using other solvents [50]. In addition, the molecular weight of chitosan influences its effectiveness as a coating agent for nanoparticles in the process of mucosal adsorption [51]. A lower molecular weight is preferable in order to enhance mucoadhesion and drug permeation [51]. Chitosan with molecular weights ranging from low (110 KDa) [49] to high (10,000 KDa) [38] have been used to coat DTX-loaded liposomes. In addition, chitosan with a molecular weight lower than 4 KDa exhibits anti-tumor effects [52]. Therefore, based on a method previously used in our laboratory for the one-step synthesis of liposomes [28], we used an ethanol injection method to fabricate our anionic liposomes coated with 1.5 KDa chitosan.

While several studies have investigated the use of DTX-loaded chitosomes in breast cancer [38,47,49], none have investigated the use of such a drug delivery system in head and neck cancer. Additionally, existing DTX-loaded chitosomes are primarily designed for oral ingestion or intravenous injection [53]. However, as mentioned above, systemic intravenous [6,7,11,12,13] leads to unwanted high toxicities and extravasation. Thus, we designed a novel chitosome formulation to benefit from recent locoregional treatment, e.g., the intra-arterial administration [14,15,16], with the aim of reducing highly toxic locoregional damage to the laryngeal mucosae.

In this study, we produced chitosomes and evaluated their applicability as nanocarriers of chemotherapy drugs for laryngeal cancer. We hypothesized that chitosomes would potentially circumvent chemotherapy drug insolubility and attenuate the adverse side effects of systemic and locoregional chemotherapy deliveries. To demonstrate the feasibility of our approach, we first developed DTX-loaded chitosomes and performed a thorough physicochemical characterization, using nanoparticle tracking analysis and zeta potential measurements, FTIR spectroscopy, and electron and fluorescence microscopies. Then, mucoadhesive studies were performed by chitosome immersion in mucin-rich dispersions. In vitro studies were further performed to evaluate the DTX therapeutic effect on human vocal fold fibroblasts (HVFFs), laryngeal squamous cell carcinomas (LSCCs), and red blood cells. 

## 2. Results and Discussion

### 2.1. DTX-Loaded Chitosomes Possessed the Expected Physical Properties

#### Size and Surface Charge Analysis

Nanoparticle tracking analysis of the non-coated and blank liposomes indicated a size of 99.4 ± 1.5 nm for the nanocarriers (Figure 2a). A similar size (107 nm to 116 nm) and spherical morphology were also reported by Paun et al. [28] when the ethanol injection method was used for the fabrication of liposomes. The addition of the chitosan coating resulted in a significant increase in size of the nanoliposomes from 99.4 ± 1.5 nm to 120 ± 3.1 nm (Figure 2a). Additionally, the polydispersity index (PDI) was below 0.2, denoting a consistent size distribution, which was similar to the 0.17 PDI (~140 nm) reported by Zafar et al. [49]. The ethanol injection method has been reported to provide more reproducible sizes in comparison to thin-film synthesis. Chitosome studies using chloroform as the organic solvent yielded nano-liposomes with PDI values ranging from 0.18 to 0.33 (~90 nm) [38] and from 0.22 to 0.41 (~240 nm) [47].

The chitosan coating was designed to create a positively charged mucoadhesive surface on the liposomes to allow electrostatic interactions with the mucosal epithelial layer of the larynx. The shift in polarity from negative to positive was noted in anionic liposomes after the chitosan coating [23,40,47]. Further, our results showed that the zeta potential shifted from −26 ± 2 mV for the blank liposomes to −8.5 mV immediately after the addition of 10 μL chitosan solution to the liposomal dispersion. The addition of more chitosan solution, i.e., from 110 μL to 170 μL (0.4 to 0.7 mg/mL of chitosan), stabilized the surface charge of blank chitosomes at 24.8 ± 2.6 mV and 28 ± 2 mV for DTX-loaded chitosomes (Figure 2a). For the zeta potential measurements, 130 μL of chitosan solution with a concentration of 0.5 mg/mL was used. The zeta value is an important parameter to take into consideration when developing nanoparticles, since it provides information about the colloidal stability and aggregation potential of nanoparticles in suspension [54]. When zeta potential values exceed 25 mV, whether positive or negative, repulsive forces are produced for better dispersion of drug-loaded nanocarriers, and thus, for more efficient delivery of their cargo [55,56]. Our zeta values for both blank liposomes and chitosomes were higher than 20 mV, which confirms the colloidal stability of our nanocarriers. 

The loading of the drug into chitosomes changes the size and charge of liposomal systems. Compared to blank liposomes (99.4 ± 1.5 nm) and blank chitosomes (120 ± 3 nm), the drug loading significantly increased the nanocarrier size (DTX-loaded liposomes = 118 ± 1.4 nm; DTX-loaded chitosomes = 130.4 ± 0.9 nm) as expected. DTX-loaded chitosomes and controls (DTX-loaded liposomes, blank chitosomes, and blank liposomes) showed a roughly spherical structure ranging from 120 nm to 150 nm in diameter according to transmission electron micrographs (Figure 2b), which are consistent with the size of these nanoparticles recorded by the nanoparticle tracking analysis. 

As expected, the addition of chitosan shifted the negative zeta potential values from −27 mV for the non-coated liposome to +28 mV for the chitosan-coated liposomes (Figure 2a). To investigate the stability of the nanoliposomes, we performed a 35-day stability evaluation on blank liposomes and chitosomes as well as their DTX-loaded versions. All groups maintained their baseline size for the first 28 days (* *p* < 0.05) (Figure 2c). However, the zeta values showed different trends across groups from day 1 to day 35. In particular, both DTX-loaded and blank chitosome groups showed a significant decrease in the magnitude of the zeta values at day 35 (Figure 2d) (* *p* < 0.05). The decrease in zeta values of the chitosomes might result from partial degradation of the chitosan coating. In contrast, the increased zeta values for the blank liposomes may have resulted from the oxidation/hydrolysis of the phospholipids membrane of the liposomes. The blank, non-coated liposomes may have become fused over an extended time, resulting in an increase in the zeta value [57].

Chitosomes’ physical properties, such as sizes of 100 to 200 nm, are key for rational nanocarrier design to enhance permeability and the retention effect [18]. When the lipoids S75 and S100 were used, the chitosan coating was shown to increase the size of non-coated liposomes up to 18% [47]. In our case, the change in size of the non-coated liposomes was about 10%, which may be due to the variation in anionic liposomal formulation. Nevertheless, this size is within the nanocarrier diameter ranges (100–200 nm) for proper circulating performance in the tumor vasculature [58]. The DTX-loading induced increases in size and in charge are consistent with those previously reported for chitosomes loaded with docetaxel (100–150 nm in size [43] and 29.8 ± 2.4 mV in surface charge [23]). Since in some of our investigations as described in Section 2.3.2, we used fluorescently labelled liposomes and chitosomes, we also measured their sizes and zeta potentials, but the difference was not significant compared to unlabelled nanocarriers.

### 2.2. Addition of a Chitosan Coating Showed Improved Mucoadhesiveness and Drug Release Profile of Liposomes

#### 2.2.1. FTIR Spectroscopy

FTIR analysis was performed to characterize the chemical composition of the chitosomes. The chitosan coating on DTX-loaded liposomes was confirmed by the presence of the peaks at 753 cm^−1^ and 893 cm^−1^ corresponding to N-H bending and to the glycosidic C-O-C stretching of chitosan, respectively [59], which were absent in the blank liposome controls (Figure 3, Table 1). The blank liposome spectrum was characterized by a peak at 1740 cm^−1^, representative of the C=O stretching of the ester bond of the lipid components, which links the head group to the fatty acid tail of the phospholipids. Lipid-related peaks were also detected at 2800 cm^−1^ corresponding to CH_2_ symmetric stretching, and at 3400 cm^−1^ for O-H and N-H stretching. The peak at 528 cm^−1^ was associated with the P-O asymmetrical bending of the PO_4_^−3^ molecule found in phospholipids [60]. The decrease in these peak intensities also confirmed a successful chitosan coating on the liposomes. The DTX encapsulation in blank liposomes was also confirmed by the presence of a peak at 710 cm^−1^, a fingerprint of the N-H bending of the benzamide in the drug [61,62] (Figure 3, Table 1).

#### 2.2.2. Mucoadhesive Studies

The results from the turbidity and surface charge tests confirmed the mucoadhesive properties of the chitosan coating (Figure 4a–c). The chitosomes were exposed to mucin 1 originating from bovine submaxillary glands for up to 3 h. The submandibular gland mucosal environment has sero-mucinous properties similar to that of the larynx [63]. The turbidity results showed an increased interaction between the mucin suspension and the chitosomes in comparison to the non-coated group recorded as an increased absorbance after 2 h (*** *p* < 0.001) and 3 h (**** *p* < 0.0001) (Figure 4b). A similar turbidity trend was reported by Yamazoe et al. [64] with their system consisting of elcatonin-loaded chitosomes and elcatonin-loaded liposomes.

The surface charge also decreased from 29.1 ± 3.3 mV to −16 ± 4.20 mV (**** *p* < 0.0001) in the DTX-loaded chitosomes after mucin exposure (Figure 4c). This positive-to-negative switch can be explained by hydrophobic and hydrogen bonding as well as the electrostatic and ionic interactions between the cationic chitosan coating and the anionic mucin suspension [64,65]. No significant changes in turbidity and surface charge were noted in the non-coated and blank liposomes, assumably owning to the anionic and repulsive interactions between the anionic liposomal surface and mucin dispersion.

The mucoadhesive behavior of the DTX-loaded chitosomes may relate to increased interstitial retention in mucin-dominant tumors, such as those found in the larynx [66]. The interactions between chitosomes and mucins are crucial for the intended use of these liposomal-based nanocarriers in head and neck cancers. Mucins, which are glycosylated proteins produced by epithelial cells to form mucus, are highly secreted by head and neck squamous cell carcinomas [67,68] such as LSCCs [66]. In particular, overexpression of Mucin-1 was associated with a worse prognosis in head and neck cancer [66,67,68].

#### 2.2.3. DTX Entrapment and Release Studies

The DTX entrapment efficiency in chitosomes was 82.6 ± 3.6% with regard to the DTX standardization curve (Figure 5a,b). As the chitosan coating was performed after the simultaneous fabrication and DTX loading of the liposomes, no significant difference was found in drug entrapment efficiency compared to non-coated liposomes (79.9 ± 3.1%). This entrapment efficiency was similar to those reported in other water-insoluble drug encapsulations, such as copper (II) diethyldithiocarbamate [28].

The DTX solution release of ~80% from the dialysis membrane was similar to that reported by Sinhg et al. (2019) after 12 h [44] (Figure 5c). Our results confirmed that the DTX release was slower from the chitosomes than from the non-coated liposomes (Figure 5c,d). Both liposome groups showed a first-order DTX release profile consistent with previous reports [69]. As opposed to non-coated anionic liposomes, the DTX release from chitosomes was expected to be prolonged due to the coating, which provides an external physical barrier enveloping the liposomes [47,70].

Our results also revealed the effect of the physiological environment on drug retention. Exposing the liposomes and chitosomes to a release medium disrupted the liposomal bilayer, which resulted in DTX escape from the nanocarriers. The incorporation of DSPC, cholesterol, and DSPE into the liposomal formulation seemed to stabilize the liposomal membrane in terms of DTX retention and release kinetics [71]. In addition, the DTX entrapment within the liposomal membrane might also help stabilize the bilayer by occluding the pores of the liposomal membrane. An increase in drug retention of about ~12% for the coated group was observed compared to non-coated liposomes (18%) at 12 h (Figure 5c). An increased drug retention of ~11% after 7 h was noted for DTX-loaded liposomes coated with Eudragit, which is a cationic methacrylic-acid-based polymer, compared to the non-coated liposomes [70]. Similarly, the DTX release from drug-loaded chitosomes was 20% lower after 24 h at the physiological pH of 7 [38]. Longer time points were not analyzed in these studies compared to ours where we investigated DTX released up to 28 days (Figure 5d). Furthermore, less than 40% of DTX was retained in the liposomal formulations after 72 h (Figure 5d). This finding is consistent with a study where a formulation with similar phospholipid constituents was tested on breast cancer cells, and a DTX retention greater than 40% was obtained after 72 h [72]. Nevertheless, our results indicate that the therapeutic window of our lipid nanocarriers would be up to 3 days after their local administration in the laryngeal mucosae of patients.

### 2.3. Docetaxel-Loaded Chitosomes Showed Higher Toxicity to Cancer Cells Than Healthy Stromal and Blood Cells

#### 2.3.1. HVFF and LSCC Viability

The in vitro cell viability analysis conducted with both liposomal and chitosomal nanocarriers without DXT showed no noticeable cytotoxicity to HVFFs and LSCCs for up to 3 days of exposure (Figure 6a–c). The quantitative MTT assay confirmed no significance difference in cell viability for HVFFs and LSCCs in the absence of nanocarriers (Figure 6d). Using non-treated cells as a control group and setting their viability at 100% (*p* > 0.5) [73], we found that >95% of the HVFFs and LSCCs remained viable after exposure to the nanocarriers, confirming the biocompatibility of chitosomes.

#### 2.3.2. Chitosome Uptake by LSCCs

Chitosome uptake by LSCCs was observed during the first 4 h of exposure (Figure 7), a timeframe very similar to those for other cell lines such as gastric and endothelial cells [43]. The EGFR staining of the LSCC surface showed an accumulation of nanocarriers within the LSCCs, which serves as a spatial reference for nanocarrier internalization (Figure 7a). The colocalization of chitosomes with lysosomes/endosomes was also observed in LSCCs (Figure 7b), indicating that the nanocarriers in the endosomal compartment would disintegrate within its acidic environment. Overall, the cationic coating from chitosan seemed to result in more accumulation of nanocarriers in cancer cells. The electrostatic interaction of cationic nanocarriers with anionic cellular membranes may contribute to inducing endocytosis [74]. Additionally, the ionisable/cationic chitosan coating on anionic liposomes may provide an acid-dependent permeability for the diffusion of DTX after liposomal uptake by cells, leading to more effective drug release [38].

#### 2.3.3. Docetaxel-Loaded Chitosomes Effectively Reduced LSCC and HVFF Colony Formation

From the LIVE/DEAD staining, LSCC and HVFF colony formation was reduced with increasing DTX concentration up to 10 µM after 3 days of culture, in comparison to the untreated confluent control (Figure 6c). To further verify the cytotoxicity of DTX alone on HVFFs and LSCCs, an MTT assay was performed which showed a decreasing trend in cell viability with increasing the DTX dose from 100 nM to 10 μM (Figure 8a–c). Based on the IC50 (half maximal inhibitory drug concentration) calculated from the dose–response curve (Figure 8c), 1 µM (10^−6^) of DTX dose was sufficient to provide the desired therapeutic effect on these cells after a 3-day DTX exposure. As such, 1 µM DTX was used for subsequent cytotoxic analyses.

We further evaluated DTX cytotoxicity on LSCCs and HVFFs for seven days. Increased cell death was observed for the LSCCs and HVFFs exposed to both DTX-loaded groups (Figure 9a) in comparison to non-treated controls (Figure 6c). Consequently, the quantitative decreasing trend in cell viability was also noted in cell-based MTT and supernatant-based LDH assays among all groups (Figure 9c). After day 3, the LSCC viability of the DTX-loaded chitosome group was ~38% viability, whereas the controls DTX alone and DTX-loaded liposomes exhibited ~50% and ~44% cell viability, respectively. At day 7, the DTX-loaded chitosome group showed a significant difference in cancer cell death compared to DTX-loaded liposomes with an ~8% increase (* *p* < 0.05) and to the DTX alone group with a ~17% increase (**** *p* < 0.0001). However, at day 7, HVFF viability remained above 20% after exposure to the three treatments.

Cancer cells are known to form colonies especially during metastasis [75]. The biological activity of DTX-loaded chitosomes was further validated via a colony formation analysis, which consisted of macroscopic staining using crystal violet (Figure 10a–c) and microscopic immunostaining of the cytoskeleton with β-tubulin III (ALEXA488/TUBIII) and counterstaining (DAPI) (Figure 10d–f). Such observations can be used to support the results described in Figure 9. The macroscopic crystal violet (Figure 10a,b) and microscopic immunostaining (Figure 10d,e) experiments confirmed that DTX-loaded chitosomes inhibited LSCC growth and proliferation after 7 days of exposure. In detail, the absorbance at 590 nm measurements showed less clonogenic activity after a 7-day treatment of DTX-loaded chitosomes (* *p* < 0.05) compared to controls (Figure 10c). Nuclei counting also showed a decrease in cell numbers of LSCC colonies after DTX-loaded chitosome exposure (* *p* < 0.05) compared to controls (Figure 10f). Further, the morphology of liposomes, i.e., round shape and ~100 nm size, is known to favor cellular intake and drug internalization [17,30,31,32,33,76]. To verify the liposomes’ capacity for sustained drug release, the cytotoxicity effects were compared between the groups of DTX-loaded liposomes and DTX-alone over an extended 7 days of exposure. As expected, the DTX-loaded anionic liposome group showed significantly more cell death (i.e., cytotoxicity) (* *p* < 0.05, Figure 10) over the course of the study, which confirmed the benefit of liposomes in anti-cancer therapeutics.

#### 2.3.4. Chitosomes Did Not Induce Hemolysis of Human Red Blood Cells

For intra-arterial locoregional delivery, the nanocarriers will inevitably interact with blood cells in the bloodstream. It is thus imperative to assess the cytotoxicity of chitosome on blood cells in addition to tumor and stromal cells. Hemolysis is defined by decomposition of the red blood cell membrane and hemoglobin release resulting in a red tint to the solution. The released hemoglobin after oxidation becomes methemoglobin and then cyanmethemoglobin [65]. After 3 h of exposure of red blood cells to a DTX concentration of 1 μM in the DTX-loaded nanocarrier, the observed hemolysis was less than 5%, indicating a non-hemolytic effect of DTX-loaded chitosomes and other liposomal formulations of these nanocarriers (Figure 11). According to the American Society for Testing and Materials International (ASTM 2013), for a value greater than 5%, the compound is considered hemolytic to red blood cells [65]. Altogether, these results prove that our drug delivery system is non-hemolytic and suitable for intra-arterial administration. However, compared to non-coated lipid nanocarriers, chitosan-coated lipid nanocarriers have been reported to provide increased antiangiogenic effects which shown in a chick embryo chorioallantoic membrane assay [49]. We anticipate a similar antiangiogenic behavior for our proposed chitosome nanocarriers.

Chitosan coatings on drug-loaded liposomes have been shown to increase cytotoxicity in terms of cancer cell death [23,24,38,39,43,47]. This is likely because the chitosan coating improves the mucoadhesive and permeability and retention properties of nanoliposomes. This would explain the increased cytotoxic effect of DTX-loaded chitosomes towards LSCCs compared to DTX-loaded liposomes. Alongside the desired in vitro cytotoxicity towards LSCCs without compromising the viability of stromal cells, the chitosomes showed no hemolytic effects on LSCCs and HVFFs.

Despite the promising in vitro results, further preclinical investigation of the DTX-loaded chitosomes is still needed for the translational pipeline of this drug delivery system. For instance, in vivo experiments for laryngeal cancer may not be fully representative models because subcutaneous injection is used to induce carcinogenesis in flanks [77] or armpits [78] of mice instead of the laryngeal anatomical site, which may cause the subject to suffocate. For this reason, the implementation of advanced in vitro models that closely mimic in vivo conditions, such as those found in the head and neck, are required to evaluate the translation potential of chitosomes prior to designing in vivo experiments [1].

## 3. Materials and Methods

### 3.1. Materials

Docetaxel (cat. # PHR1883), cholesterol (cat. # C8667), 1,2-distearoyl-sn-glycero-3-phosphocholine (DSPC, cat. # 850365P), 1,2-distearoyl-sn-glycero-3-phosphoethanolamine-N-[methoxy(polyethylene glycol)-2000]-ammonium salt (DSPE-PEG2000, cat. # 880120P), Liss Rhod PE (cat. # 810150P) and mucin from bovine submaxillary gland (cat. # M3895) from bovine submaxillary gland, human laryngeal cancer cell line (LSCC, cat. # UM-SCC-17A), FITC, UV-transparent 96-well plates, T-75 flasks, Dulbecco’s modified Eagle’s medium (DMEM), fetal bovine serum (FBS), non-essential amino acids, penicillin/streptomycin, and cyanmethemoglobin/Drabkin’s reagent were purchased from Millipore-Sigma (Burlington, MA, USA). Chitosan (cat. # 150597) with a molecular weight of 1526.464 g/mol was purchased from MP Biomedicals (Irvine, CA, USA). Dialysis membranes of 3.5–5 kDa (cat. # 131204T) and 12–14 kDa (cat. # 132703T) were purchased from Spectrum Chemical Mfg. Corp. (Gardena, CA, USA). The human vocal fold fibroblast immortalized cell line (HVFF) and 8-chamber culture slides (cat. # 154534 Lab-Tek^®^II) were obtained from the University of Wisconsin-Madison (Madison, WI, USA) and Fisher Scientific (Ottawa, ON, Canada), respectively. Human red blood cells (cat. # IWB3ALS40ML) were purchased from Innovation Research (Novi, MI, USA) and were used for the hemolysis studies. The LIVE/DEAD staining kit (cat. # L3224), MTT assay kit (cat. # V13154), and Blue LysoTracker (cat. # L7525), and cell dissociation reagent TrypLE (cat. # 12604013) were purchased from Thermo Fisher Scientific (Waltham, MA, USA). Crystal violet (cat. # C581-25) was purchased from Fisher Scientific (Ottawa, ON, Canada). The LDH assay kit (cat. # ab65393), ALEXA647/EGFR (cat. # ab192982), ALEXA488/TUBIII (cat. # ab195879), and DAPI (cat. # ab228549) from Abcam (Cambridge, UK) were used to visualize the cell membrane, cytoskeleton, and nucleus, respectively. Trypan blue (cat. # 10702404) from Invitrogen (Waltham, MA, USA) was used for cell counting.

### 3.2. Fabrication of Liposomes

Liposomal drug encapsulation is influenced by the phase transition temperatures (T_m_) of the constituent phospholipids [79]. Phospholipids have a specific T_m_ [71,79]. A T_m_ over the physiological temperature may be logically preferred for prolonged liposomal stabilization once inside the body. For instance, the 1,2-distearoyl-sn-glycero-3-phosphocholine (DSPC) has a T_m_ ≈ 55 °C [71]. This T_m_ above the physiological temperature was found to have a greater drug encapsulation efficacy in comparison to a T_m_ lower than 37 °C [79]. In addition, PEG grafting is a common approach in designing liposomal-based drug delivery systems. The reason behind the strategy for using PEGylated lipids, such as amphiphilic polymers consisting of hydrophilic 1,2-distearoyl-sn-glycero-3-phosphoethanolamine-N-[methoxy(polyethylene glycol)-2000] (DSPE-PEG2000), is that they have a T_m_ ≈ 74 °C [71] ensuring longer liposomal stabilization in the body [79].

In contrast to previously reported DTX-loaded chitosome synthesis protocols [23,24,38,39,43,47,49,53], our methodology modified a one-step liposome synthesis method that was previously used in our laboratory [28] and to synthesize DTX-loaded liposomes [72,80]. Briefly, DSPC, DSPE-PEG2000, and cholesterol with a molar ratio of 2/0.18/1 were dissolved in 2 mL of 100% ethanol by gentle stirring. For fluorescence microscopy visualization, 0.1 mg of Liss Rhod PE (orange colour) was added to the liposomal formulation.

For DTX loading, the formulation was heated at 50 °C for 5 min. Ethanol injection was performed by pouring the lipid dispersion into a flask with 300 mL of MilliQ water while stirring at 1200 rpm. After 5 min of stirring, the formulation was filtered using a coarse paper filter, and the solvent was removed by rotatory evaporation. The mixtures were dialyzed using 12–14 kDa membranes at room temperature for 30 min against 0.5% Tween 80 (pH 7.4) to remove free DTX from the DTX-loaded liposomes [81]. The liposomal dispersion was then stored at 4 °C until use.

### 3.3. Fabrication of Chitosomes

Firstly, 1.8 g of chitosan and 3 mL of 0.06 M HCL were added to 300 mL of MilliQ water (6 mg/mL) [82]. The solution was stirred in a fume hood until the chitosan was completely dissolved. The chitosan solution was then filtered, and the pH was adjusted to 5 using 1 M NaHCO_3_ to protonate the amine groups in chitosan and obtain polycationic chitosan. The solution was again filtered by two successive filtration processes using 0.45 µm and 0.22 µm filters.

A volume of 0 µL to 170 µL chitosan solution was then added to 1.5 mL of either the DTX-loaded liposomal or blank liposomal dispersion and sonicated in a water bath for 15 min (Figure 12). In detail, nine increasing concentrations of 0.4, 0.12, 0.2, 0.28, 0.36, 0.44, 0.52, 0.6, to 0.68 mg/mL of chitosan solution were used. Similar to non-coated liposomes loaded with DTX, the mixtures were dialyzed using 3.5–5 kDa membranes at room temperature for 30 min against 0.5% Tween 80 (pH 7.4) to separate the free chitosan from the liposomes. To obtain the fluorescent version of the chitosomes, Liss Rhod PE-liposomes and FITC-tagged chitosan (green colour) were used to coat the liposomes. The chitosome formulation was then stored at 4 °C until use or at 37 °C for stability studies.

### 3.4. Size and Zeta Potential Analyses of Chitosomes

A volume of 10 µL of the liposomal dispersion was placed in a 1.5 mL Eppendorf tube containing 1 mL of MilliQ water (pH = 7). After vortexing for 30 s, 1 mL of the dispersion (1:5000) was analysed by nanoparticle tracking analysis (NTA) using a Nanosight NS300 system from (Malvern Instruments Ltd., Worcestershire, UK) using a 640 nm laser, at T = 25 °C. The polydispersity index (PDI) was calculated using Equation (1).
(1)$$PDI={\left(\frac{standard\;deviation}{mean\;size}\right)}^{2}$$

For zeta potential measurements, 1.5 mL of the chitosome formulation (1:10) in distilled water was analyzed using a ZetaPALS zeta potential analyzer (Brookhaven Instruments Corp., Holtsville, NY, USA). The stability of chitosome formulation was assessed at 37 °C in a 5% CO_2_ atmosphere for freshly prepared chitosomes and after 5 weeks of storage.

### 3.5. Transmission Electron Microscopy of Chitosomes

The morphology of the DTX-loaded chitosomes was examined by transmission electron microscopy (TEM). A volume of 10 µL of the DTX-loaded chitosome and liposome samples was placed on a carbon-coated copper grid. The negative staining was performed on the formed thin film of samples on the grid by adding 2% filtered uranyl acetate (*w*/*v*) (pH 7.00). TEM images of the samples were acquired using a Tecnai G2 F20 TEM (Hillsboro, OR, USA) at a voltage of 120 kV.

### 3.6. FTIR Characterization of Chitosomes

The FTIR spectra were acquired in transmission mode using a Spectrum II (PerkinElmer Inc., Shelton, CT, USA) spectrophotometer equipped with an Attenuated Total Reflection module, single bounce diamond crystal, and Spectrum software (https://www.mcgill.ca/mc2/instrumentation/thermal-analysis-and-spectroscopy/ftir-spectrum-ii, accessed on 1 June 2023). Standard FTIR settings such as room temperature, LiTaO_3_ (lithium tantalate) MIR detector, unique humidity shield design (OpticsGuardTM) system, Pearl Liquid Analyser—liquid transmission accessory, and ZnSe 200 µm windows were used for acquiring the spectra. The spectral resolution was at 4 cm^−1^ within a 4000–600 cm^−1^ range with background clearance. A total of 128 scans were averaged for each tested sample. Baseline correction and atmospheric compensation was applied to all spectra.

### 3.7. Mucoadhesive Behavior of Chitosomes

Mucin-1 from bovine submaxillary glands was used to assess the mucoadhesive behavior of chitosomes compared to control blank liposomes. The mucin powder was suspended and stirred in 100 mM acetate buffer at a 0.5 mg/mL concentration at pH 4.4 overnight [64,65]. The mucin suspension was then centrifuged at 13,500 rpm for 20 min at 4 °C. The supernatant was filtered through a 0.22 µm filter. A 1.5 mL aliquot of the mucin suspension was placed in a centrifuge tube, and then chitosomes and liposomes (1:10) were added into each corresponding individual tube and vortexed. The suspensions were then incubated for 1, 2, and 3 h at 37 °C. The turbidity of the suspension was measured using the Spectramax i3 plate reader (Molecular Devices, San Jose, CA, USA) at 500 nm along with recording the zeta potential for changes in surface charge after another round of washes with water, centrifugation, and filtration steps.

### 3.8. Drug Entrapment Efficiency and Release of Chitosomes

The absorbance of DTX was first calibrated for different DTX concentrations using the Spectramax i3 plate reader at λ = 230 nm. To quantify the DTX entrapment efficiency, the nanocarrier supernatant containing unloaded drug was collected and diluted at various concentrations in 1 × PBS and centrifuged at 2000 RCF for 5 min to remove aggregates [28]. The samples were placed in UV-transparent 96-well plates and the absorbance of DTX at λ = 230 nm was recorded using the above-mentioned plate reader. The entrapment efficiency (EE) was then calculated according to Equation (2).
(2)$$EE\left(\%\right)=\frac{concentration\;of\;DTX\;detected\;in\;release\;medium(µg/mL)}{concentration\;of\;DTX\;added\;initially\;into\;chitosomes\;(µg/mL)\;}\times 100$$

For the DTX release kinetic analysis, 2 mL of the liposomal suspensions and 1 mg DTX solution were dialyzed using 12–14 kDa membranes and poured into 400 mL of release medium containing 0.5% Tween 80 (pH 7.4) at 37 °C [44]. The release profile of DTX was assessed after 0.5, 1, 2, 4, and 12 h as well as after 1, 1.5, 3, 7, 14, and 28 days at 37 °C. The cumulative release of DTX from chitosomes was then calculated as the percentage of DTX released at each time point compared to the amount encapsulated initially.

### 3.9. HVFFs and LSCCs Culture Protocol

Two cell lines were used for this study: (i) a non-chemoresistant human laryngeal cancer cell line (LSCC) isolated from a primary laryngeal carcinoma located at the supraglottis in the T2 or T3 stage [83,84] of a 48-year-old female patient who did not benefit from radiotherapy; and (ii) a human vocal fold fibroblast (HVFF) immortalized cell line [85] (between passages 3 and 5) representing stromal cells in the laryngeal tumor. Both cell lines were grown in LSCC complete media consisting of high glucose DMEM, 10% FBS, 1% non-essential amino acids and 1% penicillin/streptomycin in a humidified atmosphere of 5% CO_2_ at 37 °C. After reaching 70–80% confluency in T-75 flasks at passage 3 to 5, the cells were cultured in fresh FBS-free media for 1 day to synchronize their cell cycles, and were then harvested using TrypLE for 5 min. After adding LSCC media, the cells were counted using a hemocytometer before being centrifuged at 900 rpm for 5 min. The medium was discarded and the cells were resuspended in fresh LSCC media with a working concentration of 1 × 10^6^ cells/mL.

### 3.10. HVFF and LSCC Viability Analyses for Liposomes and Chitosomes

The HVFF and LSCC viability assay was performed with approximately 1 × 10^4^ cells seeded separately onto 8-chamber slides. After reaching 100% confluency (set as day 0) [86], non-drug-loaded chitosomal and control liposomal dispersions were added to the culture media at a concentration of 1/1000 [87]. At the 1- and 3-day time points, the cells were washed with 1× PBS before being stained using a LIVE/DEAD viability/cytotoxicity assay kit following the manufacturer’s instructions. The slides were incubated for 30 min in darkness at room temperature before being washed twice with 1× PBS. An inverted fluorescence microscope (Axiovert3, Zeiss, Germany) with a 10× objective was used to acquire images of cells stained with FITC (LIVE, green) and Cy3 (DEAD, red/orange). Cells were considered dead if the LIVE/DEAD staining signals overlapped [22].

MTT analysis was also carried out to obtain quantitative data on cell viability. For this assay, about 5 × 10^3^ cells were seeded into individual 96-well plates. A Spectramax i3 plate reader was used to determine the absorbance of MTT at λ = 570 nm. The percentage of cell viability was calculated using Equation (3).
(3)$$Cell\;viability\left(\%\right)=\frac{Non-treated\;control-treated\;cells}{Non-treated\;control\;}\times \;100$$

### 3.11. Chitosome Uptake by LSCCs

FITC-labelled chitosan was used to coat the Liss Rhod PE-liposomes as described in Section 2.3. Approximately 1 × 10^4^ LSCCs were seeded onto 8-chamber slides and incubated with docetaxel-loaded FITC-labelled chitosomes and the non-coated liposomal formulation (control) for up to 4 h [88] at a concentration of 1/1000 [87]. The LSCC chitosomal uptake was analyzed after 0.5 and 2 h of incubation via immunostaining following a 4 h inspection via cell tracker staining. ALEXA647/EGFR for cell membrane staining and DAPI as a counterstain along with Blue LysoTracker for lysosome staining were used to track the internalization of chitosomes and lysosomes/endosomes. The immunostaining procedure was performed following the manufacturer’s guidelines. The cells were then imaged at 40× and 63× magnifications using the Zeiss Axiover3 (Zeiss, Germany) and retrieved using Imaris version 9.5.1 Software (Bitplane, South Windsor, CT, USA).

### 3.12. DTX-Loaded Chitosome Effect on LSCCs

To determine the therapeutic effect of DTX on LSCCs, the toxic effect of 100 nM, 500 nM, 1 µM, and 10 µM DTX alone was assessed on HVFFs and LSCCs after 3 days via LIVE/DEAD staining, MTT analysis, and LDH assay. LSCCs were then exposed to DTX-loaded chitosomes for up to 7 days. Approximately 1 × 10^4^ cells were seeded onto 8-chamber slides. After reaching 100% cell confluency, 1 µM DTX-loaded liposomal or chitosomal dispersions were added to the culture media at a 1/1000 dilution [87] in each of the slide’s chambers. The addition of 1 µM DTX alone to cells was used as control. The DTX formulation cytotoxicity was then investigated at days 1, 3, 5, and 7 using the LIVE/DEAD assay. Furthermore, the MTT assay was carried out as described in Section 3.10 to obtain quantitative cell cytotoxicity data for each formulation treatment. In addition, the LDH assay was performed to further corroborate the MTT results by analyzing the supernatant following the manufacturer’s guidelines. The MTT and LDH percentages of cell viability were calculated using Equation (3).

To investigate the colony formation after exposure to DTX alone, DTX-loaded liposomes, and DTX-loaded chitosomes, LSCCs were seeded in 24-well plates at a density of 15 × 10^3^ and incubated until full confluency up to 7 days. After the treatments, the culture medium was removed, and the cells were washed twice with PBS. Then, the cells were stained with 0.1% crystal violet (in water 30%, ethanol 70%) in sterile water (0.5 mL/well) for 30 min at room temperature. After thorough washing, the colonies were analyzed via absorbance at 590 nm [89] with a Spectramax i3 plate reader. Hampering of colony formation was analyzed via immunostaining of the cytoskeleton and nuclei using an Axiovert3 microscope with a 20× objective and Cy3, ALEXA488, and DAPI filters. Images were acquired using Imaris 9.5.1 Software, and all cell nuclei (10 µm) in the entire image were counted using the spot detection algorithm and DAPI mean fluorescence intensity [90].

### 3.13. Hemolytic Effect of Chitosomes

The nanocarrier formulations at predetermined concentrations with and without DTX were added to diluted (100×) red blood cells in PBS to a final volume of 12 mL. Tween 80 was used as a positive control. The samples were mildly shaken for 1, 2, and 3 h at 37 °C and centrifuged at 800 g for 15 min. Hemolysis was evaluated by mixing 100 μL of each supernatant with 100 μL of cyanmethemoglobin/Drabkin′s reagent in a 96-well plate and reading the absorbance at 540 nm using a Spectramax i3 plate reader. The percentage of hemolysis was calculated using Equation (4).
(4)$$Hemolysis\left(\%\right)=\frac{absorbance\;sample-absorbance\;negative\;control}{absorbance\;positive\;control-absorbance\;negative\;control\;}\times \;100$$

### 3.14. Statistical Analysis

The data are reported as mean ± SE or SD of at least three experiments. The statistical significance of the differences was analyzed by two-way ANOVA and Tukey post hoc tests using GraphPad Prism version 9.5.1 for Windows (GraphPad Software, San Diego, CA, USA).

## 4. Conclusions

In this study, we fabricated DTX-loaded chitosomes with optimal physical properties for enhanced permeability and retention features. The DTX-loaded chitosomes consisted of anionic nanoliposomes and a cationic chitosan coating to provide the nanoliposomes with mucoadhesive and increased drug retention properties. The overall in vitro results suggest that chitosomes loaded with docetaxel are a promising delivery system for laryngeal cancer via intra-arterial administration. DTX-loaded chitosomes indeed showed improved anti-cancer effects on a laryngeal cancer cell line with no signs of hemolysis. Further in vitro studies are required to better understand the fate and bioactivity as well as the targetability of DTX-loaded chitosomes in the laryngeal cancer tumor microenvironment.

## Figures and Tables

**Figure 1 ijms-24-09902-f001:**
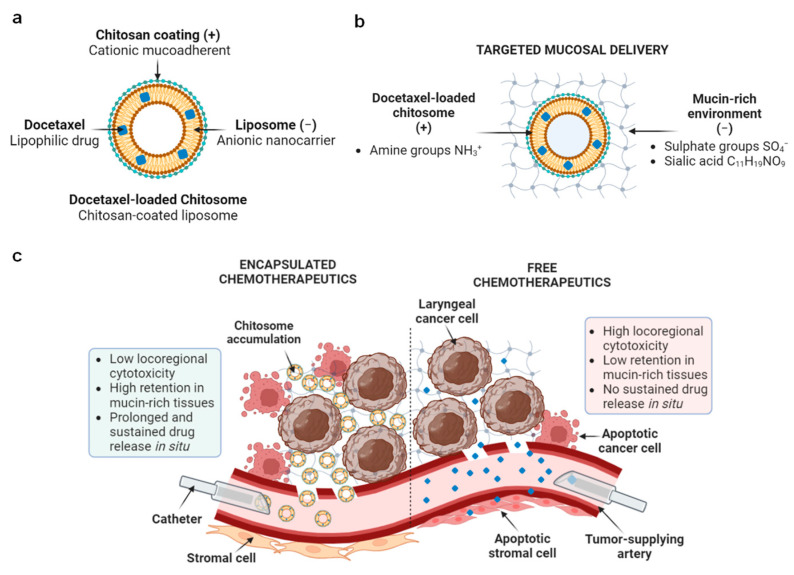
Schematic representation of chitosan-coated liposomes, namely ‘chitosomes’, as chemo drug nanocarriers. (**a**) Components of docetaxel-loaded chitosomes. (**b**) Mucin–chitosome electrostatic interactions. (**c**) Encapsulated versus free drug comparison in mucin-rich tumors. Figure created with BioRender.com accessed on 1 June 2023.

**Figure 2 ijms-24-09902-f002:**
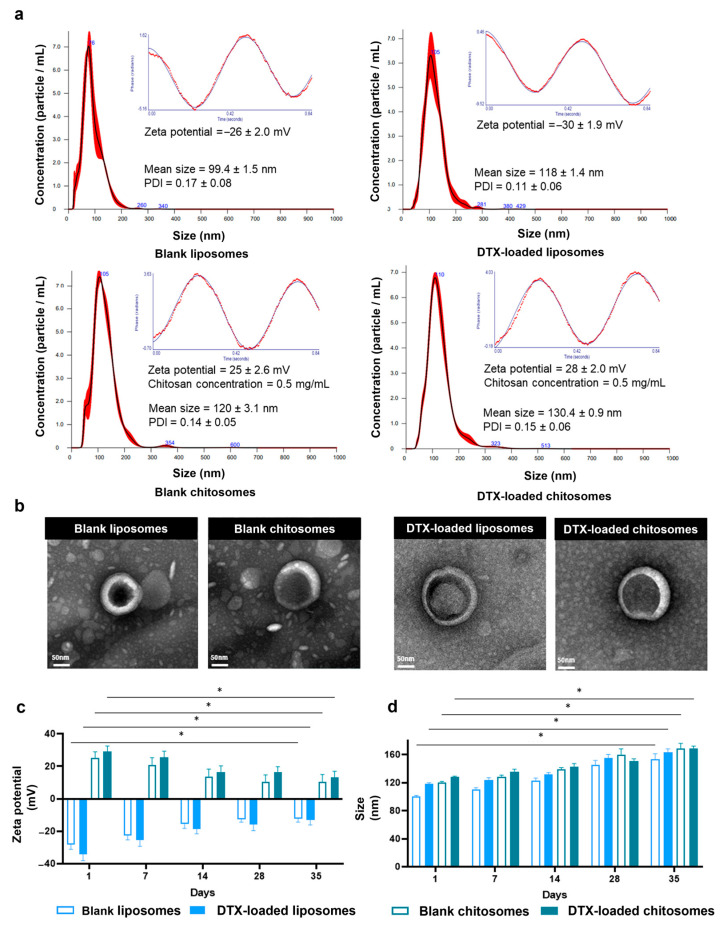
Size and charge of the blank and DTX-loaded liposomes coated with chitosan analyzed at neutral pH. (**a**) Optimized size and phase analysis light scattering plots of the blank and DTX-loaded liposomes coated with chitosan. Sample data (red lines/dots) overlays their fits (black/blue lines). Chitosan concentration was calculated using the C_1_V_1_ = C_2_V_2_ dilution formula as (6 mg/mL chitosan concentration) (10 μL chitosan volume) = C_2_ (1.5 mL liposomal suspension). (**b**) Morphology of the blank and DTX-loaded liposomes coated with chitosan via TEM. Scale bar = 50 nm. The changes in size and charge of the DTX-loaded chitosomes were significant in comparison to all blank groups. (**c**) Stability test at 37 °C of DTX-loaded liposomal formulation. (**d**) Zeta potential of the DTX-loaded non-coated and coated liposomes after 5 weeks at 37 °C and 5% CO_2_. Size and zeta potential data are reported as mean ± SE and mean ± SD, respectively. * *p* < 0.05.

**Figure 3 ijms-24-09902-f003:**
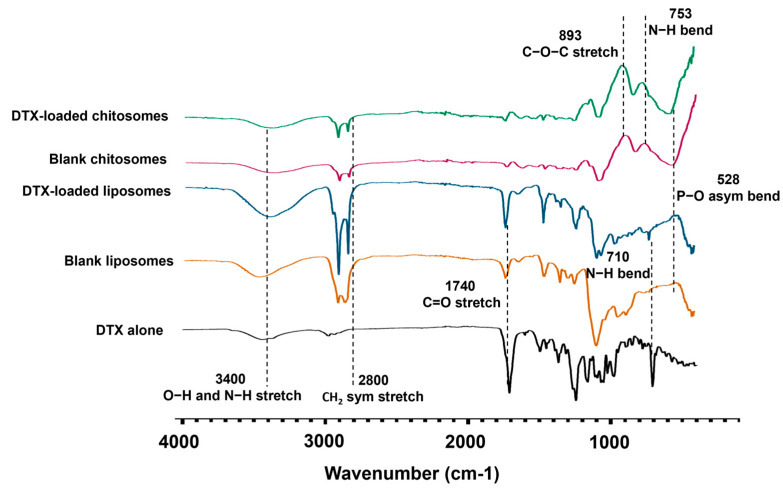
FTIR spectra of blank and DTX-loaded chitosomes. Chemical fingerprint confirmation of chitosan presence and DTX loading in the liposomes.

**Figure 4 ijms-24-09902-f004:**
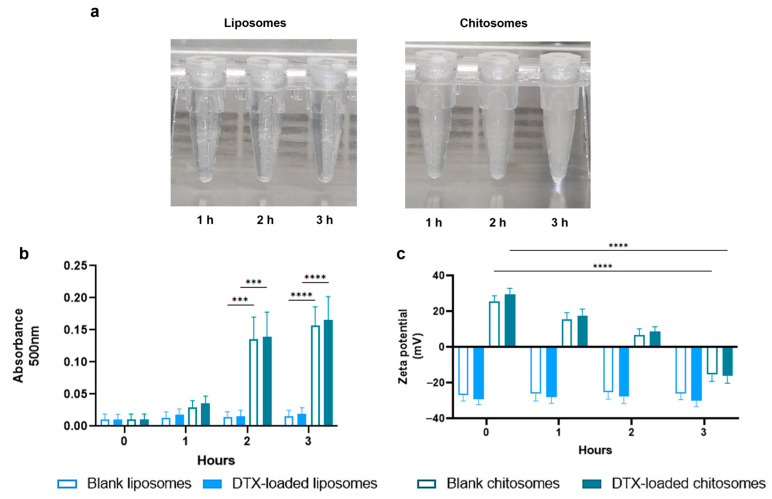
Mucoadhesive studies of DTX-loaded chitosomes. (**a**) Qualitative observation of turbidity as a result of the interaction between chitosomes or liposomes and mucin dispersion. Individual representative samples of DTX-loaded chitosomes were chosen to properly visualize the turbidity of the samples. (**b**) Turbidity as a function of mucoadhesive behavior between chitosomes or liposomes and mucin as the absorbance reading at 500 nm. (**c**) Zeta potential measurements of chitosomes or liposomes after mucin suspension exposure. Chitosomes’ mucoadhesive behavior was confirmed in mucin-containing dispersion after immersion for 3 h. *** *p* < 0.001, **** *p* < 0.0001.

**Figure 5 ijms-24-09902-f005:**
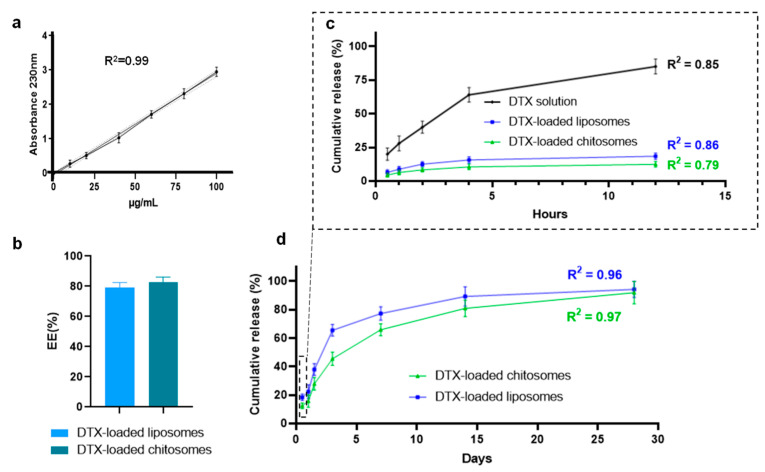
Drug release from DTX-loaded liposomes and chitosomes. (**a**) DTX standardization curve (absorbance at 230 nm). Sample data with their fit margin using a linear regression plot.(**b**) Docetaxel entrapment efficacy was about 81.9 ± 5.3% after analysing the amount of the drug released in the supernatants. Drug release profile up to (**c**) 12 h and (**d**) 28 days.

**Figure 6 ijms-24-09902-f006:**
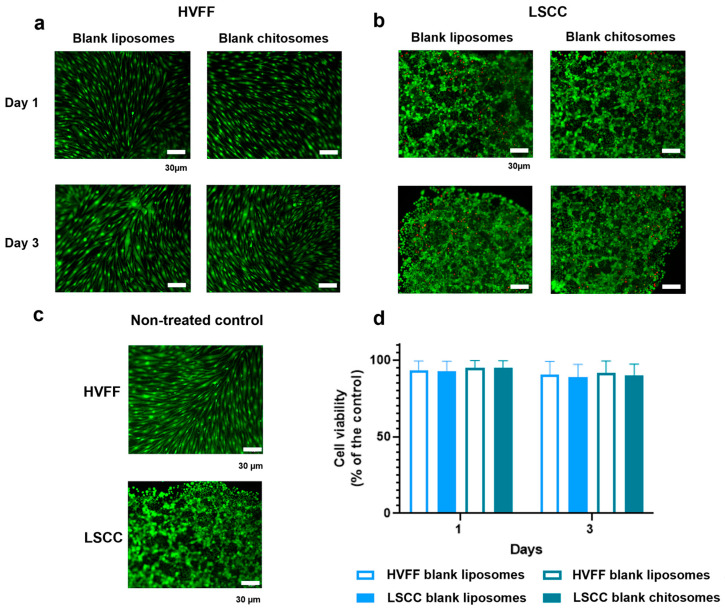
Blank liposome and chitosome effects on HVFFs and LSCCs. (**a**) Blank liposomes, (**b**) blank chitosomes, and (**c**) non-treated confluent control (Day 0). (**d**) MTT assay on the effect of blank liposomes and blank chitosomes on HVFFs and LSCCs. Non-treated HVFFs and LSCCs were set as the reference of 100% cell viability from confluency at Day 0. No significance difference was noticed among groups and culture time. Scale bar = 30 µm.

**Figure 7 ijms-24-09902-f007:**
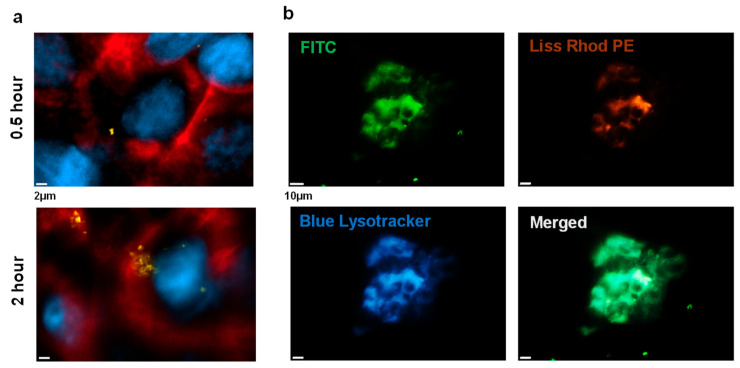
Chitosome uptake by LSCCs. (**a**) Internalization of chitosomes after 0.5 h and 2 h of treatment (bright orange, merged FITC/Liss Rhod PE). Immunofluorescence staining of cell membranes via EGFR (red) with DAPI (blue) as nuclei counterstaining was performed to visualize internal nanocarrier accumulation. Scale bar = 2 µm. (**b**) Colocalization of chitosomes and endosomes/lysosomes immediately after a 4 h chitosome exposure. The LSCC colony fluorescence shows the internalization of the FITC-labelled chitosomes (FITC, chitosan coating; orange, anionic liposome) via Blue Lysotracker (blue). Scale bar = 10 µm.

**Figure 8 ijms-24-09902-f008:**
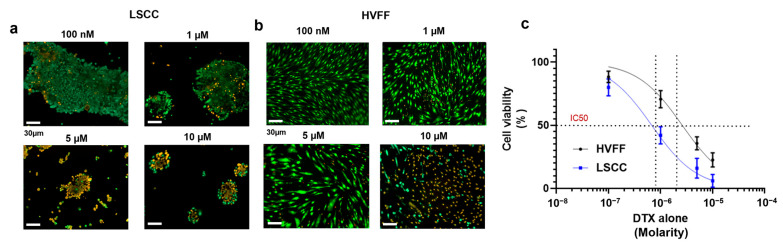
Docetaxel dose response of (**a**) LSCCs and (**b**) HVFFs. Scale bar = 30 µm. (**c**) MTT cytotoxicity assay of DTX on HVFFs and LSCCs after 3 days to determine IC50 concentration. A decreasing trend in LSCC and HVFF viability was noted with increasing dose of DTX. Green = live cells. Bright orange = dead cells.

**Figure 9 ijms-24-09902-f009:**
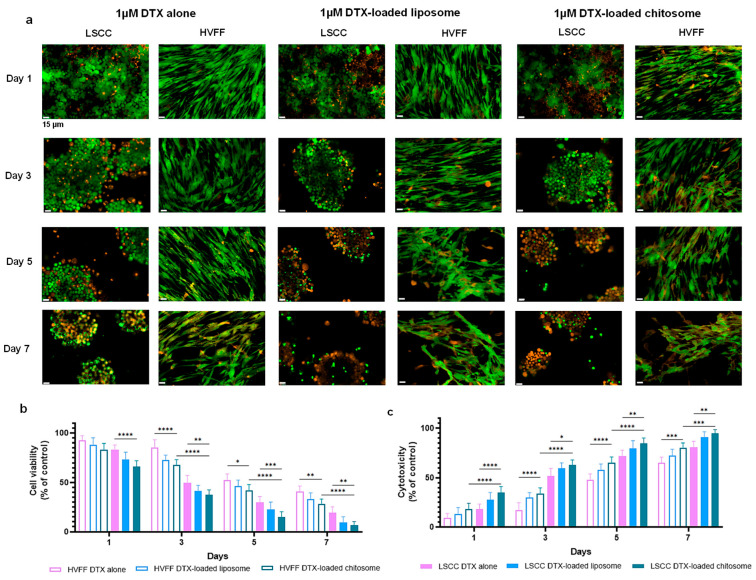
DTX-loaded chitosome effects on the viability of LSCCs and HVFFs. (**a**) Therapeutic exposure effects on LSCCs and HVFFs up to 7 days via LIVE/DEAD staining where live cells (green) dead cells (bright orange) are displayed. Scale bar = 15 µm. (**b**) MTT assay and (**c**) LDH assay of the quantitative effects up to 7 days of DTX exposure on LSCCs and HVFFs. Cell viability was reduced in the DTX alone and DTX encapsulated groups after the 7 days. * *p* < 0.05, ** *p* < 0.01, *** *p* < 0.001, **** *p* < 0.0001.

**Figure 10 ijms-24-09902-f010:**
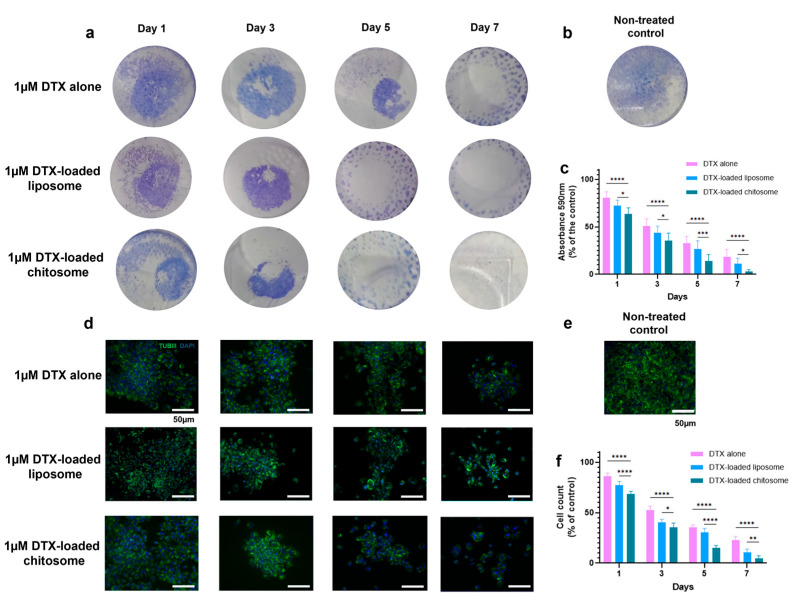
DTX-loaded chitosome effects on the colony formation of LSCCs. (**a**,**b**) Clonogenic assay via crystal violet staining. (**c**) Quantitative measurement (absorbance at 590 nm) of the crystal violet staining for clonogenic assessment. (**d**,**e**) Colony cytoskeleton observation via spot detection algorithm in DAPI channel. (**f**) A decreasing trend in cancer colony size was observed following exposure to DTX alone and DTX encapsulated groups for up to 7 days. * *p* < 0.05, ** *p* < 0.01, *** *p* < 0.001, **** *p* < 0.0001.

**Figure 11 ijms-24-09902-f011:**
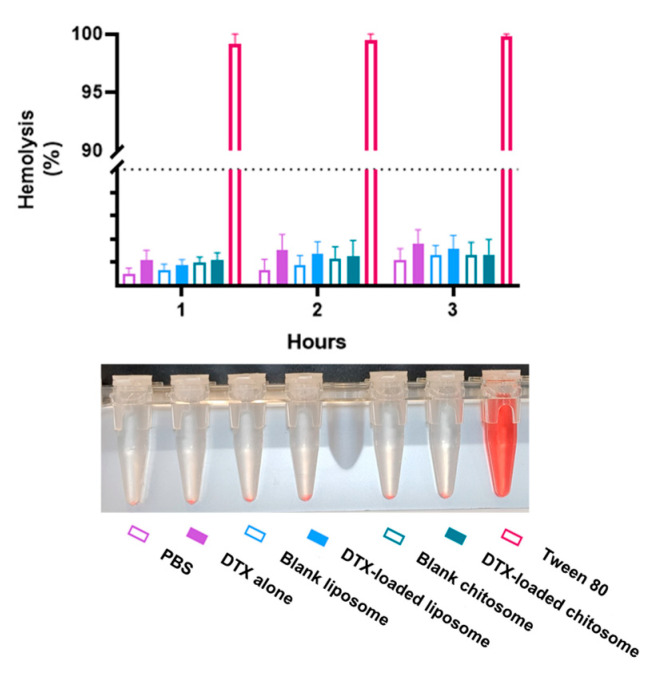
Hemolytic analysis of exposure to DTX-loaded chitosomes for up to 3 h. The hemolytic response was measured by the intensity of the red color in the assay tubes. Hemolysis of red blood cells was less than 5% indicating non-hemolytic effects of either the DTX-loaded or blank liposomal and chitosomal formulations.

**Figure 12 ijms-24-09902-f012:**
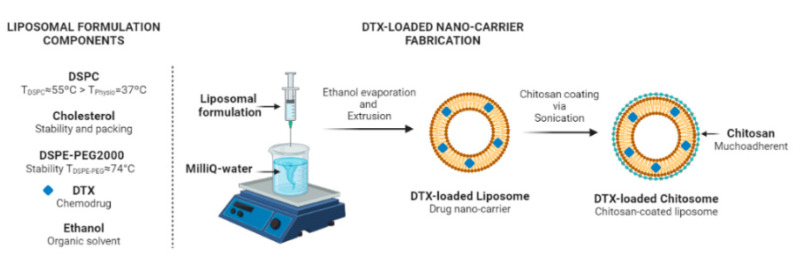
Schematic representation of DTX-loaded liposome and chitosome fabrication. Figure created with BioRender.com accessed on 1 June 2023.

**Table 1 ijms-24-09902-t001:** Identification of characteristic peaks of chitosomes compared to their constituent (controls) in their respective FTIR spectra.

Wavenumber (cm^−1^)	Vibrational Mode	Biomolecular Attributions
528	P-O asymmetrical bending	Phospholipids (PO_4_^−3^ molecule)
710	Benzamide N-H bending	Docetaxel
753	N-H bending	Chitosan
893	Glycosidic C-O-C stretching	Chitosan
1740	C=O stretching	Lipids
2800	CH_2_ stretching	Lipids
3400	O-H and N-H stretching	Lipids

## Data Availability

Not applicable.

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
