# Peer review of "Chitosomes Loaded with Docetaxel as a Promising Drug Delivery System to Laryngeal Cancer Cells: An In Vitro Cytotoxic Study"

_ijms, 2023, doi:10.3390/ijms24129902_

Round 1

Reviewer 1 Report

In this study, Moya-Gracia et al. developed and further characterized a docetaxel-loaded anionic nanoliposome coated with a layer of mucoadhesive chitosan to deliver chemotherapeutic drugs. Chitosan is naturally a polycation and its charge density depends on the degree of acetylation and the pH of the medium. Chitosan oligomers are soluble in a wide pH range, from acidic to basic (i.e., physiological pH 7.4), which makes them suitable candidates for drug delivery. Furthermore, the therapeutic effect of DTX on human vocal fold fibroblasts (HVFFs), laryngeal squamous cell carcinomas (LSCCs) and red blood cells has been demonstrated in vitro.

Major concerns:

- Chitosan is used to coat the surface of nanoliposomes and is found to increase the size of ~20nm. The positive zeta potential value confirms the chitosan surface coating. I would like to know how stable and uniform the chitosan coating is. If the chitosan solution remains in the solution and no washing step is involved, how can we determine the efficiency of the coating? I assume that the chitosan is not a crosslinker that forms the outer gel layer. If chitosan crosslinks the outer surface of nanoliposomes, can it affect the stability and quantification ability? Many chemical and physical crosslinkers for chitosan have been reported (T. Jóźwiak et al. / Reactive and Functional Polymers 114, 58-74, 2017).

- The zeta potential of chitosan changes with the pH of the buffer (Monsalve et al., Nanomedicine (Lond.), 10(11), 1735-1750, 2015).  The pH of the chitosan solution is adjusted to 5. At what pH is the zeta potential measured and why is pH 5 chosen?

- It was found that the surface charge of DTX-loaded chitosomes decreased after mucin exposure. Could this be due to the dissolution of chitosan? It seems that at this time the pH of the medium changes, which could lead to the dissolution of the chitosan layer due to the basic conditions.

- It is interesting to see that ionizable/cationic chitosan coating on anionic liposomes may provide acid-dependent permeability for the diffusion of DTX leading to effective drug release. What can be expected from this behavior in an in vivo study? What parameters and observations can be considered? 

- In some previous reports cationic polymer-based DTX drug delivery system has been developed, e.g., Wang et al., 11 (2),133-14, 2016. What can one take advantage of using this system for drug delivery applications and clinical trials?

- It was hard to follow Figure 6, why there is no stable imaging? The quality of imaging needs to be improved.

Minor comments:

1. In the introduction section, line 46, “docetaxel (DTX) is one common chemotherapy drug” needs to be more explicit. It seems there is something missing here.

2.  The resolutions of Figures 1, 2, 3, 4,8, 10, and 11 need to be improved. Some of the text seems to be a blur. In Figure 2, size distribution analysis, it seems the authors used system generated pdf file. It needs to be plotted with high resolution.

N/A

Author Response

  1. Chitosan is used to coat the surface of nanoliposomes and is found to increase the size of ~20nm. The positive zeta potential value confirms the chitosan surface coating. I would like to know how stable and uniform the chitosan coating is. If the chitosan solution remains in the solution and no washing step is involved, how can we determine the efficiency of the coating? I assume that the chitosan is not a crosslinker that forms the outer gel layer. If chitosan crosslinks the outer surface of nanoliposomes, can it affect the stability and quantification ability? Many chemical and physical crosslinkers for chitosan have been reported (T. Jóźwiak et al. / Reactive and Functional Polymers 114, 58-74, 2017).

We thank the reviewer for this comment. The chitosan coating on the anionic liposomes was performed without any chemical and physical cross-linking agent. Such coating occurred due to electrostatic interactions between the cationic nature of chitosan (pH 5 to 6) and the negatively charged surface of the liposomes. We performed wash steps to get rid of free chitosan in the solution. If the washing steps were not performed, we might not be able to assess the effective coating and the readings might be biased.

  1. The zeta potential of chitosan changes with the pH of the buffer (Monsalve et al., Nanomedicine (Lond.), 10(11), 1735-1750, 2015).  The pH of the chitosan solution is adjusted to 5. At what pH is the zeta potential measured and why is pH 5 chosen?

Thanks for the question. The zeta potential studies were performed at neutral pH 7 as we used distilled water as the suspension medium. During the coating process, the chitosan solution was adjusted to pH of 5, in order to allow the protonation of chitosan and its interaction with the negatively charged liposomes. This information is now added to the ‘Materials and Methods”, Section 3.4.

  1. It was found that the surface charge of DTX-loaded chitosomes decreased after mucin exposure. Could this be due to the dissolution of chitosan? It seems that at this time the pH of the medium changes, which could lead to the dissolution of the chitosan layer due to the basic conditions.

We thank the reviewer for bringing this to our attention. We believe that the surface charge shift is due to electrostatic interactions between anionic mucin macromolecules and the chitosomes, rather than the dissolution of chitosan layer. In principle, the chitosan coating should be stable at the neutral pH at which the zeta potential of the samples is measured. It should also be noted that the mucin pH is about 6–7 (Flemstrom, 1987) and thus the chitosan may remain stable in such a pH environment.

  1. It is interesting to see that ionizable/cationic chitosan coating on anionic liposomes may provide acid-dependent permeability for the diffusion of DTX leading to effective drug release. What can be expected from this behavior in an in vivo study? What parameters and observations can be considered? 

This is indeed a very interesting observation. We believe that such a scenario can occur in mucin-rich environments in vivo as discussed in #3. To verify related hypothesis in vivo, tumor weigh loss can be used to assess changes in the volume and weight of the tumor in animal models. Concerning patients, imagen-based analysis as MRI and PET scans are longitudinally performed before and after drug administration to monitor changes in tumor size.

  1. “Liposome solution” may not be the right term. All the nanoparticles including liposomes should be described as dispersion rather than a solution.

We have replaced the “Lipid solution” by “Lipid dispersion” in Line 465.

  1. In some previous reports cationic polymer-based DTX drug delivery system has been developed, e.g., Wang et al., 11 (2),133-14, 2016. What can one take advantage of using this system for drug delivery applications and clinical trials?

We thank the reviewer for this comment. We took advantage of the mucoadhesive behavior of chitosan, which makes it a good fit as a bioadhesive for mucin-rich mucosal tissues like the larynx and upper airway.  

  1. It was hard to follow Figure 6, why there is no stable imaging? The quality of imaging needs to be improved.

We apologize for the ambiguity of Figure 6. We have now provided an improved quality of this figure and for all the figures in the manuscript.

  1. In the introduction section, line 46, “docetaxel (DTX) is one common chemotherapy drug” needs to be more explicit. It seems there is something missing here.

We now revised the sentence (Line 47) as follows: “docetaxel (DTX) is one common chemotherapy drug in the treatment of head and neck cancer [8–10]

  1. The resolutions of Figures 1, 2, 3, 4,8, 10, and 11 need to be improved. Some of the text seems to be a blur. In Figure 2, size distribution analysis, it seems the authors used system generated pdf file. It needs to be plotted with high resolution.

We apologized for the suboptimal image quality. We have now enhanced the resolution of all figures within what the journal allows in the manuscript.

Reviewer 2 Report

In this manuscript, authors synthesized docetaxel-loaded chitosan-coated liposomes (DTX-loaded chitosomes) to improve the efficacy of treatment in head and neck cancer. Indeed, a correct diagnosis of head and neck cancer is always delayed by a lacking of striking cardinal symptoms in clinical practice. The prognosis is poor. Therefore, this work is noteworthy for improving the efficacy of therapy.

I have a few questions.

1. Docetaxel-loaded chitosan-coated liposomes (DTX-loaded chitosomes) have been explored in many studies. For instance,

DOI: 10.3390/molecules24020250

DOI: 10.1016/j.colsurfb.2019.110603

DOI: 10.1007/s12668-022-01053-2

DOI: 10.1007/s13346-020-00779-4

DOI: 10.4333/KPS.2010.40.2.085

DOI: 10.1016/j.biopha.2011.09.010

What distinguishes the chitosome in this manuscript from previously published?

2. In Figure 3

a. Please confirm the labels in Figure 3a to be correct. There were no obvious peaks at 753 cm-1 and 893 cm-1 for DTX-loaded chitosomes (green) and Blank chitosomes (purple).

b. Authors can remove Figure 3b or change this figure into a table. It is not mainstream to express a table as a figure.

c. The structure diagram of the docetaxel is not necessary. Otherwise, please show the structure diagram of phospholipids, docetaxel, chitosan, and lipids, as mentioned in Figure 3b.

d. Both DSPC and DSPE-PEG are phospholipids. Please explain why author further distinguished between phospholipids and lipids in Figure 3b.

3. Figure 4

Figure 4a is not clear for reading. Please provide a new one.

4. In Figure 5

a. Figure 5a (standard curve) can be provided as supplementary.

b. For Figures 5a, 5c, and 5d, please express the figure either as a connected scatter plot only or a scatter plot with a trend line (like Figure 8C).

5. Figure 6

a. Please confirm the figure legend to be correct. Authors wrote [Blank chitosome and DTX alone effects on HVFFs and LSCCs]. However, no DTX was used in this experiment.

b. Please add the experimental time for Figure 6c.

6. Figure 7

a. The cell morphology in the Bright field looked abnormal (more like debris or aggregation). Please provide other images that clearly demonstrate the fluorescent signals inside the LSCCs. (Like Figure 7c)

b. Since the size of fluorescent signals is approximately 100 μm, it was not a single cell. However, it is hard to tell whether chitosomes overlap with the lysosomes under this magnification. In addition, it is unlikely that there will be nearly complete overlapping results since there should be different stages of endocytosis, including early endosomes, late endosomes, and lysosomes. Please provide images with higher magnification.

c. The images in Figure 7c were not clear for reading. Please provide a clear image. In addition, five images are not necessary. Two images with different magnifications are enough.

7. Figure 8

a. Please confirm the unit in Figures 8a-b. nm/μm or nM/μM?

b. Please add the unit of concentration for Figure 8c.

c. Since this experiment is more like a trial to choose a suitable concentration for the follow-up experiment, it could be provided as supplementary.

8. Figure 9 and Figure 10

1. Theoretically, PEG will reduce endocytosis, and negatively charged liposomes are not easily internalized by cells also. Why is the effect of DTX-loaded liposome better than DTX alone? An explanation of this phenomenon is believed to be expected by readers.

2. Please confirm the experimental time in Figures 9a. Day 2 or Day 3?

9. Figure 11

a. Authors mentioned that tween 80 was used as the positive control in Line 567. Why the percentage of hemolysis was not 100%?

b. The image above was not clear for reading. Please provide a clear image.

c. Please add the concentration of these samples. It is not mentioned in section 3.13, either.

10. In section 3.2

a. Please add the concentration in different experimental steps.

b. What was the condition for rotary evaporation? Different types of products can be seen in the TEM image, and some may not even be liposomes. This step may be the main cause of this phenomenon.

11. In section 3.3

Please add the concentration in different experimental steps.

12. In section 3.8

Please confirm the definition of formula (2). Drug release or entrapment efficiency?

13. In section 3.10

a. Please confirm the description of [After reaching 100% confluency] in Line 517 and Line 546. It is reasonable only if the doubling times of HVEFs and LSCCs in this study are both longer than 3-7 days.

b. What was the unit of live cells and dead cells? How did the authors calculate? For instance, Image J.

c. How to distinguish live cells and dead cells? when the cells are damaged, overlapping signals of live cells (center) and dead cells (cell periphery) can be seen as in these results.

d. Authors mentioned that the percentages of MTT and LDH were calculated using formula (3) in Line 553. [MTT and LDH percentages of cell viability were calculated using equation (3)]. How to calculate dead cells using MTT assay?

e. Data expressed with cell viability was all performed by MTT assay (Figures 6, 8, and 9). Please confirm the definition of formula (3). Live/Dead assay or MTT assay.

Author Response

  1. Docetaxel-loaded chitosan-coated liposomes (DTX-loaded chitosomes) have been explored in many studies. For instance,

DOI: 10.3390/molecules24020250 [Alshraim et al (2019)]

DOI: 10.1016/j.colsurfb.2019.110603 [Zafar et (2020)]

DOI: 10.1007/s12668-022-01053-2 [Ahmed et al (2023)]

DOI: 10.1007/s13346-020-00779-4 [Sun et al (2021)]

DOI: 10.4333/KPS.2010.40.2.085

DOI: 10.1016/j.biopha.2011.09.010

What distinguishes the chitosome in this manuscript from previously published?

We thank the reviewer for this advice. Among the recommended studies, we have selected the most recent publications, i.e., Sunet al. (2021), Alshraim et al. (2019), Zafar et al. (2020), Ahmed et al. (2023), that appear as references 38, 48, 50 and 54, respectively. We also added another recent study by Guo et al. (2021), [reference 49: 10.1016/j.colsurfb.2020.111499] to provide a more comprehensive coverage of the literature related to the work presented in our manuscript.  Even though it seems that similar concepts are used in those studies as compared to ours, the originality of our work reside overarchingly in the following aspect.

In particular, we designed a one-step liposome synthesis with subsequent chitosan coating while in the studies cited above, multiple steps were used to synthesize DTX-loaded “chitosomes”, i.e., liposomes coated with high molecular weight chitosan [Sun et al (2020), Alshraim et al (2019), Guo et al (2021), Zafar et al (2019)]. To reflect more in detail this liposomal synthesis difference, we added the following paragraph in Introduction, beginning on Line 100:

Multiple steps are implemented to synthesize DTX-loaded “chitosomes”, i.e., liposomes coated with chitosan [38,48–50]. Lipid components are first dissolved in organic solvents as ethanol [50] or chloroform [38,48,49] with an extra step to load DTX. The use of ethanol as organic solvent in liposome synthesis has been reported to increase reproducibility of liposome particle size and polydispersity index compared to those using other solvents [51]. In addition, the molecular weight of chitosan influences its effectiveness as a coating agent for nanoparticles in the process of mucosal adsorption [52]. A lower molecular weight is preferable in order to enhance mucoadhesion and drug permeation [52]. Chitosan with molecular weights ranging from low (110 KDa) [50] to high (10,000 KDa) [38] have been used to coat DTX-loaded liposomes. In addition, chitosan molecular weight lower than 4 KDa exhibits anti-tumor effects [53].Therefore, based on a method previously used in our laboratory for one-step synthesis of liposomes [28], we thus used an ethanol injection method to fabricate our anionic liposomes coated with 1.5 KDa chitosan.”

The other major difference between our work and the previous ones is the application of chitosomes and its future administration method. Our drug delivery system is expected to take advantage of the recent intra-arterial administration in the head and neck for locoregional treatment that could reduce drug toxicity and damage to surrounding healthy tissue in the upper airway. The following paragraph is added to the introduction, beginning on Line 114:

“While several studies have investigated the use of DTX-loaded chitosome in breast cancer [38,48,50], none have investigated the use of such a drug delivery system in head and neck cancer. This has impacted the evaluation of the chitosomes for use in head and neck cancer. Existing chitosome research has primarily focused on the evaluation of chitosomes containing DTX for oral ingestion or intravenous injection [54]. However, as mentioned above, systemic intravenous [6,7,11–13] leads to unwanted high toxicities and extravasation. Therefore, we designed a novel chitosome formulation to benefit from recent locoregional treatment, e.g., the intra-arterial administration [14–16], with the aim of reducing high toxic locoregional damage in the laryngeal mucosae

Lastly, most studies used chloroform thin-film synthesis to prepare chitosomes. In contrast, we used the ethanol injection method to prepare chitosomes with more uniform size (~130nm) and lower polydispersity index (<0.17). The following explanation is added to ‘Results and Discussion section’

Line 140: Also, the polydispersity index (PDI) was below 0.2 denoting the consistent size distribution, which was similar to the 0.17 PDI (~140nm) reported by Zafar et al [50]. Ethanol injection method has been reported to provide more reproducible size in comparison to thin-film synthesis. Chitosome studies using chloroform as an organic solvent yielded nano-liposomes with the PDI values ranging from 0.18 to 0.33 PDI (~90nm) [38] and from 0.22 to 0.41 PDI (~240nm) [48].”

Line 149:  “The shift in polarity from negative to positive was noted in anionic liposomes after the chitosan coating [23,40,48].

Line 194: “When the lipoid S75 and S100 were used, chitosan coating has shown to increase the size of non-coated conventional liposomes up to around 18% [48].”

Line 266: “As opposed to non-coated anionic liposomes, DTX release from chitosomes was expected to be prolonged due to the coating, in which providing an external physical barrier enveloping the liposomes [48,72].”

Line 279: “Similarly, the DTX release from drug-loaded chitosomes was 20% lower after 24 hours at physiological pH 7 [38].”

Line 317: “Also, the ionisable/cationic chitosan coating on anionic liposomes may provide an acid-dependent permeability for the diffusion of DTX entrapped after liposomal uptake by cells, leading to more effective drug release [38].

Line 393: “However, compared to non-coated lipid nanocarriers, chitosan-coated lipid nanocarriers have been reported to provide increased antiangiogenic effect as show in a chick embryo chorioallantoic membrane assay [50].”

Line 457: “In contrast to previously reported DTX-loaded chitosome synthesis [23,24,38,39,43,48,50,54], our methodology modified a one-step liposome synthesis previously used in our laboratory [28]…”

  1. In Figure 3
  2. Please confirm the labels in Figure 3a to be correct. There were no obvious peaks at 753 cm-1 and 893 cm-1 for DTX-loaded chitosomes (green) and Blank chitosomes (purple).
  3. Authors can remove Figure 3b or change this figure into a table. It is not mainstream to express a table as a figure.
  4. The structure diagram of the docetaxel is not necessary. Otherwise, please show the structure diagram of phospholipids, docetaxel, chitosan, and lipids, as mentioned in Figure 3b.
  5. Both DSPC and DSPE-PEG are phospholipids. Please explain why author further distinguished between phospholipids and lipids in Figure 3b.

a-b. We appreciate this feedback. Figure 3 is now updated. The data in figure 3b is reported as a table format now in Table 1 on Line 222.

  1. We removed the diagram of docetaxel from the Figure 3 as suggested.
  2. To distinguish between lipids and phospholipids, we made clearer that the phosphate group (PO4−3molecule) is the distinction at peak at 528 cm-1 associated with the P-O asymmetrical bending (Line 215).
  3. Figure 4

Figure 4a is not clear for reading. Please provide a new one.

We apologized for the confusion. Figure 4a is now updated. We increased the figure size to make the cloudiness of the solutions with enhanced visibility. The horizontal lines in the background are the backwall of the biosafety cabinet.

  1. In Figure 5
  2. Figure 5a (standard curve) can be provided as supplementary.
  3. For Figures 5a, 5c, and 5d, please express the figure either as a connected scatter plot only or a scatter plot with a trend line (like Figure 8C).
  4. We would like to keep the standard curve in the body of the manuscript to avoid the need for generating a supplementary information file because no other data will be provided as supplementary in this manuscript.
  5. We corrected and updated Figure 5 as suggested. We now plotted this figure as connected scatter plots.
  6. Figure 6
  7. Please confirm the figure legend to be correct. Authors wrote [Blank chitosome and DTX alone effects on HVFFs and LSCCs]. However, no DTX was used in this experiment.
  8. Please add the experimental time for Figure 6c.

We thank the reviewer for this observation. We revised Figure 6 with the experimental time Day 0 as non-treated confluent controls accordingly. The Figure legend is now read as “Blank liposome and chitosome effects on HVFFs and LSCCs. (a) Blank liposomes, (b) blank chitosomes, and (c) non-treated confluent control (Day 0). (d) MTT assay on the effect of blank liposomes and blank chitosomes on HVFFs and LSCCs. Non-treated HVFFs and LSCCs were set as the reference of 100% cell viability from confluency at Day 0. No significance difference was noticed among groups and culture time. Scale bar = 30µm”.

  1. Figure 7
  2. The cell morphology in the Bright field looked abnormal (more like debris or aggregation). Please provide other images that clearly demonstrate the fluorescent signals inside the LSCCs. (Like Figure 7c)
  3. Since the size of fluorescent signals is approximately 100 μm, it was not a single cell. However, it is hard to tell whether chitosomes overlap with the lysosomes under this magnification. In addition, it is unlikely that there will be nearly complete overlapping results since there should be different stages of endocytosis, including early endosomes, late endosomes, and lysosomes. Please provide images with higher magnification.
  4. The images in Figure 7c were not clear for reading. Please provide a clear image. In addition, five images are not necessary. Two images with different magnifications are enough.

a-c. We simplified and updated Figure 7 accordingly. Concerning the overlapping results, new immunostaining results are now added to present the drug uptake at 0.5h and 2h with higher magnification of 63x. We also improved the image quality of the lysosome tracker staining at the 4h timepoint with magnification of 40x.

  1. Figure 8
  2. Please confirm the unit in Figures 8a-b. nm/μm or nM/μM?
  3. Please add the unit of concentration for Figure 8c.
  4. Since this experiment is more like a trial to choose a suitable concentration for the follow-up experiment, it could be provided as supplementary.

a-b. We have updated Figure 8 with the unit of μM, i.e., micro-molar.

  1. Figure 8 is included to show the effect of the dose response. The reviewer is correct that these in vitro findings are important to guide follow-up experiments, for instance, in vivo studies. We felt that Figure 8 will provide meaningful and useful information for researchers who are also working on a similar topic. As such, we would like to keep Figure 8 in the body of the manuscript.
  2. Figure 9 and Figure 10
  3. Theoretically, PEG will reduce endocytosis, and negatively charged liposomes are not easily internalized by cells also. Why is the effect of DTX-loaded liposome better than DTX alone? An explanation of this phenomenon is believed to be expected by readers.
  4. Please confirm the experimental time in Figures 9a. Day 2 or Day 3?
  5. We agreed that the LIVE/DEAD and MTT results seem to contradict the idea that the PEG and liposome components in chitosomes might cause problems of cellular internalization from the theoretical perspective.

We speculated that the distinctive morphology of liposomes, i.e., round shape and ~100nm size, have assisted cellular uptake [Bai et al (2022), reference 78: 10.3390/mi13101623]. More importantly, the sustained release feature of liposomes might also play a key role in enhancing the cytotoxicity noted in Figures 9 and 10. As opposed to DTX alone, our DTX-loaded liposomes facilitated an extended drug release. This feature allowed the cancer cells exposed to DTX continuously over the 7 days of experiment period, resulting in a pronounced cytotoxicity outcome.

We tried to address this comment by adding a paragraph starting on Line 416: “Further, the morphology of liposomes, i.e., round shape and ~100 nm size, is known to favor cellular intake and drug internalization [17,30–33,78]. To verify liposome’s capacity of sustained drug release, cytotoxicity were compared between the groups of DTX-loaded liposomes and DTX-alone over an extended 7 days of experiment. As expected, the DTX-loaded anionic liposome group showed significantly more cell death (i.e., cytotoxicity) (*p<0.05, Figure 10) over the course of the study, which confirmed the benefit of liposomes in anti-cancer therapeutics.

  1. We increased the quality of Figure 9 as well as correcting the typo of Day 3.
  2. Figure 11
  3. Authors mentioned that tween 80 was used as the positive control in Line 567. Why the percentage of hemolysis was not 100%?
  4. The image above was not clear for reading. Please provide a clear image.
  5. Please add the concentration of these samples. It is not mentioned in section 3.13, either.

a-b. We revised Figure 11 with clearer images. The tween solution, which was used as positive control, resulted in 100% hemolysis. The exact readings are at 1h (99.2% ± 0.8), 2h (99.5% ± 0.5) and 3h (99.8% ± 0.2).

  1. In section 3.13, we updated the sample concentration as those predetermined in our previous experiments.
  2. In section 3.2
  3. Please add the concentration in different experimental steps.
  4. What was the condition for rotary evaporation? Different types of products can be seen in the TEM image, and some may not even be liposomes. This step may be the main cause of this phenomenon.
  5. We added the concentrations of the chitosan solution explaining the results (Line 150-156) and methodology (Lines 480-481).
  6. Rotary evaporation was used to remove the ethanol solvent from the liposomal dispersion. We followed the methodology of one-step liposome synthesis from Paun et al (2022), a previously published paper from Prof. Tabrizian’s lab.

Potential debris may be related to the washing step after rotary evaporation. Another possibility may be related to the TEM-based staining process and vacuum conditions that could affect the liposome conformation as noted in the TEM image.

  1. In section 3.3

Please add the concentration in different experimental steps.

We included the detailed of increasing concentration of chitosan solution into the liposomal dispersion on Line 480 as follows: “In detail, nine increasing concentrations from 0.4, 0.12, 0.2, 0.28, 0.36, 0.44, 0.52, 0.6 to 0.68 mg/mL of chitosan solution were used.”

  1. In section 3.8

Please confirm the definition of formula (2). Drug release or entrapment efficiency?

Equation 2 is for the calculation of entrapment efficiency. This has been corrected in the revised version on Line 540.

  1. In section 3.10
  2. Please confirm the description of [After reaching 100% confluency] in Line 517 and Line 546. It is reasonable only if the doubling times of HVEFs and LSCCs in this study are both longer than 3-7 days.
  3. What was the unit of live cells and dead cells? How did the authors calculate? For instance, Image J.
  4. How to distinguish live cells and dead cells? when the cells are damaged, overlapping signals of live cells (center) and dead cells (cell periphery) can be seen as in these results.
  5. Authors mentioned that the percentages of MTT and LDH were calculated using formula (3) in Line 553. [MTT and LDH percentages of cell viability were calculated using equation (3)]. How to calculate dead cells using MTT assay?
  6. Data expressed with cell viability was all performed by MTT assay (Figures 6, 8, and 9). Please confirm the definition of formula (3). Live/Dead assay or MTT assay.
  7. Thank you very much for this insightful comment. In this study, we did not measure the doubling time to assess cytotoxic effect on cells. The reviewer is thus correct that our MTT and LDH results in Figures 6, 8 and 9 did not reflect the doubling times of HVFFs and LSCCs as conventionally computed in cancer research.

Traditionally, doubling times as the growth rates of the populations, are calculated via the division of the natural logarithm of 2 by the exponent of growth.  Instead, we followed the approach by Resinger et al (2015) [reference 88] to use “A time-0 plate was developed at the time of drug addition to measure initial cell density and evaluate cytotoxicity” and observe the changes in cell numbers after drug treatment.

Here, we used ‘the non-treated confluent control’ at Day 0 to set the 100% reference and compare all other study groups across time points. In detail, we made the correction on the following lines:

Line 305: “ Non-treated HVFFs and LSCCs were set as the reference of 100% cell viability from confluency at Day 0.

Line 565: “After reaching 100% confluency at Day 0 [88], non-drug loaded chitosomal and control liposomal dispersions were added to the culture media at a concentration (1/1000) [89].”

  1. We used plate reader to quantify the absorbance of the MTT and LDH assays to estimate cell counts. Thus, the units are representative of the absorbance emitted by the analyzed cells.
  2. We considered cells as dead if LIVE/DEAD staining signals overlapped as previously done in Prof. Li-Jessen’s lab, c.f., Coburn et al (2020), reference 22.

d-e. We did not measure the number of cells in the cytotoxic studies. We clarified and corrected the formula in corresponding section.

Reviewer 3 Report

The manuscript entitled “Chitosomes loaded with docetaxel as a promising drug delivery system to laryngeal cancer cells: An in vitro cytotoxic study” is well organized and the data presented are interesting.

In order to improve the quality of the manuscript, please take into account a few observations.

 1. Introduction. Please present more clearly the originality of the study.

2. Results and Discussion

At page 5, lines 138-142, the authors said that: “Values greater than 20 mV of zeta potential either positive or negative, convey repulsion forces for better dispersion of drug-loaded nanocarriers, and thus, for more effcient delivery of their cargo [50]. Our zeta values for both blank liposomes and chitosomes were higher than 20 mV, which confirms the colloidal stability of our nanocarriers.”

In general, particles having ZP values higher than 25 mV (negative or positive) are recognized as being highly stable [Shnoudeh, A.J., Hamad, I., Abdo, R.W., Qadumii, L., Jaber, A.Y., Surchi, H.S., Alkelany, S.Z. 2019. Synthesis, Characterization, and Applications of Metal Nanoparticles. Biomater. Bionanotechnol. 527–612. doi:10.1016/b978-0-12-814427-5.00015-9.]. Please clarify this aspect in the text of the manuscript.

3.2.1. Section

-Please change the title of 3.2.1. section from “Chemical fingerprint analysis”  with an appropriate title, for example: FTIR spectroscopy

-Please change figure 3. The characteristic peaks are not clearly visible. Also, add the characteristic peaks on the figure.

3.6. Section

Please describe the operating mode better in section 3.6

3.8. Section

It is not clear! What is the amount of drug (DTX) used to load the liposomes?

In the equation 2 specify which is the measure unit used. µg/ml? mg/ml?

Author Response

We thank the reviewer for this advice. Among the recommended studies, we have selected the most recent publications, i.e., Sun et al. (2021), Alshraim et al. (2019), Zafar et al. (2020), Ahmed et al. (2023), that appear as references 38, 48, 50 and 54, respectively. We also added another recent study by Guo et al. (2021), [reference 49: 10.1016/j.colsurfb.2020.111499] to provide a more comprehensive coverage of the literature related to the work presented in our manuscript.  Even though it seems that similar concepts are used in those studies as compared to ours, the originality of our work reside overarchingly in the following aspect.

In particular, we designed a one-step liposome synthesis with subsequent chitosan coating while in the studies cited above, multiple steps were used to synthesize DTX-loaded “chitosomes”, i.e., liposomes coated with high molecular weight chitosan [Sun et al (2020), Alshraim et al (2019), Guo et al (2021), Zafar et al (2019)]. To reflect more in detail this liposomal synthesis difference, we added the following paragraph in Introduction, beginning on Line 100:

Multiple steps are implemented to synthesize DTX-loaded “chitosomes”, i.e., liposomes coated with chitosan [38,48–50]. Lipid components are first dissolved in organic solvents as ethanol [50] or chloroform [38,48,49] with an extra step to load DTX. The use of ethanol as organic solvent in liposome synthesis has been reported to increase reproducibility of liposome particle size and polydispersity index compared to those using other solvents [51]. In addition, the molecular weight of chitosan influences its effectiveness as a coating agent for nanoparticles in the process of mucosal adsorption [52]. A lower molecular weight is preferable in order to enhance mucoadhesion and drug permeation [52]. Chitosan with molecular weights ranging from low (110 KDa) [50] to high (10,000 KDa) [38] have been used to coat DTX-loaded liposomes. In addition, chitosan molecular weight lower than 4 KDa exhibits anti-tumor effects [53].Therefore, based on a method previously used in our laboratory for one-step synthesis of liposomes [28], we thus used an ethanol injection method to fabricate our anionic liposomes coated with 1.5 KDa chitosan.”

The other major difference between our work and the previous ones is the application of chitosomes and its future administration method. Our drug delivery system is expected to take advantage of the recent intra-arterial administration in the head and neck for locoregional treatment that could reduce drug toxicity and damage to surrounding healthy tissue in the upper airway. The following paragraph is added to the introduction, beginning on Line 114:

“While several studies have investigated the use of DTX-loaded chitosome in breast cancer [38,48,50], none have investigated the use of such a drug delivery system in head and neck cancer. This has impacted the evaluation of the chitosomes for use in head and neck cancer. Existing chitosome research has primarily focused on the evaluation of chitosomes containing DTX for oral ingestion or intravenous injection [54]. However, as mentioned above, systemic intravenous [6,7,11–13] leads to unwanted high toxicities and extravasation. Therefore, we designed a novel chitosome formulation to benefit from recent locoregional treatment, e.g., the intra-arterial administration [14–16], with the aim of reducing high toxic locoregional damage in the laryngeal mucosae

Lastly, most studies used chloroform thin-film synthesis to prepare chitosomes. In contrast, we used the ethanol injection method to prepare chitosomes with more uniform size (~130nm) and lower polydispersity index (<0.17). The following explanation is added to ‘Results and Discussion section’

Line 140: Also, the polydispersity index (PDI) was below 0.2 denoting the consistent size distribution, which was similar to the 0.17 PDI (~140nm) reported by Zafar et al [50]. Ethanol injection method has been reported to provide more reproducible size in comparison to thin-film synthesis. Chitosome studies using chloroform as an organic solvent yielded nano-liposomes with the PDI values ranging from 0.18 to 0.33 PDI (~90nm) [38] and from 0.22 to 0.41 PDI (~240nm) [48].”

Line 149:  “The shift in polarity from negative to positive was noted in anionic liposomes after the chitosan coating [23,40,48].

Line 194: “When the lipoid S75 and S100 were used, chitosan coating has shown to increase the size of non-coated conventional liposomes up to around 18% [48].”

Line 266: “As opposed to non-coated anionic liposomes, DTX release from chitosomes was expected to be prolonged due to the coating, in which providing an external physical barrier enveloping the liposomes [48,72].”

Line 279: “Similarly, the DTX release from drug-loaded chitosomes was 20% lower after 24 hours at physiological pH 7 [38].”

Line 317: “Also, the ionisable/cationic chitosan coating on anionic liposomes may provide an acid-dependent permeability for the diffusion of DTX entrapped after liposomal uptake by cells, leading to more effective drug release [38].

Line 393: “However, compared to non-coated lipid nanocarriers, chitosan-coated lipid nanocarriers have been reported to provide increased antiangiogenic effect as show in a chick embryo chorioallantoic membrane assay [50].”

Line 457: “In contrast to previously reported DTX-loaded chitosome synthesis [23,24,38,39,43,48,50,54], our methodology modified a one-step liposome synthesis previously used in our laboratory [28]…”

  1. Results and Discussion. At page 5, lines 138-142, the authors said that: “Values greater than 20 mV of zeta potential either positive or negative, convey repulsion forces for better dispersion of drug-loaded nanocarriers, and thus, for more efficient delivery of their cargo [50]. Our zeta values for both blank liposomes and chitosomes were higher than 20 mV, which confirms the colloidal stability of our nanocarriers.” In general, particles having ZP values higher than 25 mV (negative or positive) are recognized as being highly stable [Shnoudeh, A.J., Hamad, I., Abdo, R.W., Qadumii, L., Jaber, A.Y., Surchi, H.S., Alkelany, S.Z. 2019. Synthesis, Characterization, and Applications of Metal Nanoparticles. Biomater. Bionanotechnol. 527–612. doi:10.1016/b978-0-12-814427-5.00015-9.]. Please clarify this aspect in the text of the manuscript.

We thank the reviewer for this comment. We now included the study from Shnoudeh et al (2019) as reference 57. We also clarified the claim in Line 158: “When zeta potential values exceed 25 mV, whether positive or negative, they provide repulsive forces for better dispersion of drug-loaded nanocarriers, and thus, for more efficient delivery of their cargo [56,57].”

  1. Section 3.2.1 -Please change the title of 3.2.1. section from “Chemical fingerprint analysis” with an appropriate title, for example: FTIR spectroscopy -Please change figure 3. The characteristic peaks are not clearly visible. Also, add the characteristic peaks on the figure.

We updated the title of the subsection 3.2.1 as suggested by reviewer. We also improved the image quality of Figure 3.

  1. 3.6. Section- Please describe the operating mode better in section 3.6

We thank the reviewer for this observation and updated the FTIR methodology in Section 3.6.

FTIR spectra was acquired in transmission mode using a Spectrum II (PerkinElmer Inc, USA) spectrophotometer equipped with an Attenuated Total Reflection module, single bounce diamond crystal, and Spectrum software. Standard FTIR settings such as room-temperature, LiTaO3 (lithium tantalate) MIR detector, unique humidity shield design (OpticsGuardTM) system, Pearl Liquid Analyser – liquid transmission accessory, and ZnSe 200 µm windows were used for acquiring the spectra. The spectral resolution was at 4 cm−1 within a 4000–600 cm−1 range with background clearance. A total 128 scans were averaged for each testing sample. Baseline correction and atmospheric compensation was applied to all spectra.”

  1. 3.8 Section - It is not clear! What is the amount of drug (DTX) used to load the liposomes? In the equation 2 specify which is the measure unit used. µg/ml? mg/ml?

The amount of drug loaded into the nanoliposomes is 1 mg/ml, by following the guideline of Singh et al. (2015), and Paun et al. (2022) [from Prof Tabrizian’s lab one-step liposome synthesis protocol]. The information is now included in Section 3.8. The corresponding calibration curve is in Figure 5a.

Round 2

Reviewer 1 Report

I appreciate the authors' efforts to address my concerns expressed in the first round of reviews. The quality of the manuscript has improved significantly, and it is almost ready to be published in IJMS. However, I think it would be of great benefit to the community working in this area if the authors could clarify some points in the manuscript.

It is mentioned that the zeta potential was measured at pH 7. The zeta potential of chitosan becomes more positive at acidic pH, which is due to the NH2 group (Ramanert et al., Nanoscale Research Letters, 8, 512, 2013; Monsalve et al., Nanomedicine (Lond.), 10(11), 1735-1750, 2015).

- During the coating process, the pH was adjusted to 5 to protonate the chitosan. I wonder why the zeta potential of chitosan is measured at 7 and not also at pH 5 and why the authors did not use the same pH as the wash medium to ensure consistency? For example, if the coating is done at pH 5 and the pH of the solution is changed to 7, there will be less charge-based interaction, causing the coating to dissolve. No crosslinker is used to form a gel layer. However, when chitosan with crosslinker becomes a gel at acidic pH, it destabilizes at higher pH. There is no evidence that a change in pH has no effect on the chitosan layer. If the coating is not significant, it will lead to a decrease in the efficiency of the developed system. On the other hand, poly-l-lysine is a positive polymer and its zeta potential remains positive even at neural to basic pH conditions, which could be a promising candidate for coating in buffer exchange systems (Ma et al., Chemistry, 7;15(47):13135-40, 2009. Why did the authors focus on chitosan? Please motivate the reader.

- Figure 2a: The font size of the x-axis and y-axis is difficult to read. Please increase the resolution of the inset figures. Also, include the pH value in the caption.

- The text in the figure on the first page also needs to be resized.

Author Response

  1. I appreciate the authors' efforts to address my concerns expressed in the first round of reviews. The quality of the manuscript has improved significantly, and it is almost ready to be published in IJMS. However, I think it would be of great benefit to the community working in this area if the authors could clarify some points in the manuscript.

It is mentioned that the zeta potential was measured at pH 7. The zeta potential of chitosan becomes more positive at acidic pH, which is due to the NH2 group (Ramanert et al., Nanoscale Research Letters, 8, 512, 2013; Monsalve et al., Nanomedicine (Lond.), 10(11), 1735-1750, 2015).

During the coating process, the pH was adjusted to 5 to protonate the chitosan. I wonder why the zeta potential of chitosan is measured at 7 and not also at pH 5 and why the authors did not use the same pH as the wash medium to ensure consistency? For example, if the coating is done at pH 5 and the pH of the solution is changed to 7, there will be less charge-based interaction, causing the coating to dissolve. No crosslinker is used to form a gel layer. However, when chitosan with crosslinker becomes a gel at acidic pH, it destabilizes at higher pH. There is no evidence that a change in pH has no effect on the chitosan layer. If the coating is not significant, it will lead to a decrease in the efficiency of the developed system. On the other hand, poly-l-lysine is a positive polymer and its zeta potential remains positive even at neural to basic pH conditions, which could be a promising candidate for coating in buffer exchange systems (Ma et al., Chemistry, 7;15(47):13135-40, 2009. Why did the authors focus on chitosan? Please motivate the reader.

We appreciate the reviewer's comment. The application of a chitosan coating onto the anionic liposomes was achieved without employing any chemical or physical cross-linking agent. The coating occurred as a result of electrostatic interactions between the positively charged properties of chitosan (at pH 5 to 6) and the negatively charged surface of the liposomes.

We primed the reader about the use of chitosan as a mucoadhesive agent on liposomes. Chitosan-coated liposomes electrostatically interact with negatively charged molecules, sulphate groups (SO4-), found in oral mucin and schematically pictured in Figure 1b-c.  We also increased the font size of this Figure as advised in the next comment. Please find the corresponding revision from Line 88 to 100.

Line 88: “The application of a chitosan coating on anionic nanoliposomes can serve as a mucoadhesive agent, enhancing the ability to retain drugs and increase their uptake by target tumors [23,24,38,40,42,45]. For instance, chitosan-coated nanocarriers have shown to attach to anionic glycoproteins in oral mucin  [45,46] via electrostatic interactions, and as a result, increased the retention rate of nanocarriers intratumorally (Figure 1b). This mucoadhesive feature is particularly desired for mucin-dominant mucosae like those commonly found in head and neck tumors. Chitosan-coated nanocarriers also showed to lower their aggregation in the blood and the liver [47], as well as increase transcellular and paracellular drug transport for prolonged drug release [45].”

Line 97: “Schematic representation of chitosan-coated liposomes, namely ‘chitosomes’, as chemo drug nanocarriers. (a) Components of docetaxel-loaded chitosomes. (b) Mucin-chitosome electrostatic interactions. (c) Encapsulated versus free drugs comparison in mucin-rich tumors. Figure created with BioRender.com

Regarding the use of neutral pH to perform zeta potential measurements, the zeta potential is strongly influenced not only by the pH of the solution but also by the concentration and type of salts dissolved in the solution as explained by Lowry et al (2016) (DOI: 10.1039/C6EN00136J).

Zeta potential is generally measured at neutral pH, especially for nanoparticles intended to be used as drug delivery systems  as it is an indication of colloidal stability (Samimi et al., https://doi.org/10.1016/B978-0-12-814031-4.00003-9). More specifically to chitosmes, we followed the protocol for the zeta potential measurement published by Bonechi et al (2021) (DOI: 10.3390/app112412071) and Badran et al (2022) (DOI: 10.3390/pharmaceutics14040826).

- Figure 2a: The font size of the x-axis and y-axis is difficult to read. Please increase the resolution of the inset figures. Also, include the pH value in the caption.

We thank the reviewer for this comment. We updated Figure 2 by increasing the font size and including the pH at the caption.

Figure 2 caption: “Figure 2. Size and charge of the blank and DTX-loaded liposomes coated with chitosan analyzed at neutral pH = 7. (a) Optimized size and phase analysis light scattering plots of the blank and DTX-loaded liposomes coated with chitosan. Chitosan concentration was calculated using the C1V1 = C2V2 dilution formula as (6mg/mL chitosan concentration)(10μL chitosan volume) = C2(1.5mL liposomal suspension). (b) Morphology of the blank and DTX-loaded liposomes coated with chitosan via TEM. Scale bar = 50nm. The changes in size and charge of the DTX-loaded chitosomes were significant in comparison to all blank groups (c) Stability test at 37°C of DTX-loaded liposomal formulation. (d) Zeta potential of the DTX-loaded non-coated and coated liposomes after 5 weeks under 37°C and 5% CO2. Size and zeta potential data are reported as mean ± SE and mean ± SD, respectively.”

- The text in the figure on the first page also needs to be resized.

We updated Figure 1 by increasing the font size as suggested.

Reviewer 2 Report

The authors have addressed previous comments. A few minor revisions are still needed.

In Figure 3

a. The authors wrote [The chitosan coating on DTX-loaded liposomes was confirmed by the presence of the peaks at 753 cm-1 and 893 cm-1 corresponding to N-H bending and to the glycosidic C-O-C stretching of chitosan, respectively [61], which were absent in the blank liposome controls (Figure 3, Table 1).]. However, there were no obvious peaks at 753 cm-1 and 893 cm-1 for DTX-loaded chitosomes (green line) and Blank chitosomes (purple line), still. Please confirm the labels are correct.

b. Please set the range of the X-axis from 4000 cm-1 to 600 cm-1, from left to right.

c. Please adjust the figure scale by extending the length of the X-axis or shortening the length of the Y-axis.

In Figure 5a

Please express the standard curve as a scatter plot with a trend line since the quantification of DTX needs to be calculated through a linear trend line.

Author Response

  1. The authors have addressed previous comments. A few minor revisions are still needed.

In Figure 3

  1. The authors wrote [The chitosan coating on DTX-loaded liposomes was confirmed by the presence of the peaks at 753 cm-1 and 893 cm-1 corresponding to N-H bending and to the glycosidic C-O-C stretching of chitosan, respectively [61], which were absent in the blank liposome controls (Figure 3, Table 1).]. However, there were no obvious peaks at 753 cm-1 and 893 cm-1 for DTX-loaded chitosomes (green line) and Blank chitosomes (purple line), still. Please confirm the labels are correct.
  2. Please set the range of the X-axis from 4000 cm-1 to 600 cm-1, from left to right.
  3. Please adjust the figure scale by extending the length of the X-axis or shortening the length of the Y-axis.

a-c. Thank you for bringing to our attention on this unclarity in the FTIR spectra (Figure 3). Per the advice, we changed the X-axis from left to right as enlarged the spectra to increase the visibility of peaks at 753 cm-1 and 893 cm-1 in the corresponding chitosan-related groups, i.e., green line = DTX-loaded chitosome group and purple line = Blank chitosome group.

In Figure 5a

Please express the standard curve as a scatter plot with a trend line since the quantification of DTX needs to be calculated through a linear trend line.

We appreciate this observation from the reviewer. We updated Figure 5a as recommended.